https://doi.org/10.1038/s41467-019-10023-4　　**OPEN**

# Akt and STAT5 mediate naïve human CD4+ T-cell early metabolic response to TCR stimulation

Nicholas Jones [1], Emma E. Vincent [2,3], James G. Cronin [1], Silvia Panetti [1], Megan Chambers[1], Sean R. Holm[1], Sian E. Owens [1], Nigel J. Francis [1], David K. Finlay [4,5] & Catherine A. Thornton[1]

Metabolic pathways that regulate T-cell function show promise as therapeutic targets in diverse diseases. Here, we show that at rest cultured human effector memory and central memory CD4+ T-cells have elevated levels of glycolysis and oxidative phosphorylation (OXPHOS), in comparison to naïve T-cells. Despite having low resting metabolic rates, naive T-cells respond to TCR stimulation with robust and rapid increases in glycolysis and OXPHOS. This early metabolic switch requires Akt activity to support increased rates of glycolysis and STAT5 activity for amino acid biosynthesis and TCA cycle anaplerosis. Importantly, both STAT5 inhibition and disruption of TCA cycle anaplerosis are associated with reduced IL-2 production, demonstrating the functional importance of this early metabolic program. Our results define STAT5 as a key node in modulating the early metabolic program following activation in naive CD4+ T-cells and in turn provide greater understanding of how cellular metabolism shapes T-cell responses.

[1] Institute of Life Science, Swansea University Medical School, Swansea SA2 8PP, UK. [2] MRC Integrative Epidemiology Unit, University of Bristol, Oakfield House, Bristol BS8 2BN, UK. [3] Cellular and Molecular Medicine, University of Bristol, Biomedical Sciences Building, Bristol BS8 1TD, UK. [4] School of Biochemistry and Immunology, Trinity Biomedical Sciences Institute, Trinity College Dublin, 152–160 Pearce Street, Dublin, Ireland. [5] School of Pharmacy and Pharmaceutical Sciences, Trinity Biomedical Sciences Institute, Trinity College Dublin, 152–160 Pearce Street, Dublin, Ireland. Correspondence and requests for materials should be addressed to C.A.T. (email: c.a.thornton@swansea.ac.uk)

T-cell memory is a fundamental feature of the adaptive immune response, and enhanced responsiveness of memory cells is important for immunological protection. Metabolic pathways (such as glycolysis and glutaminolysis) are integrally linked to all aspects of immune responses, including development, differentiation and immune effector function[1–3]. It is also becoming apparent that individual metabolites play important modulatory roles in the T-cell response[4,5].

Work in human CD8+ T-cells has revealed a metabolic basis for enhanced responses of CD8+ effector memory (EM) T-cells upon antigen restimulation[6]. However, while it is clear that activation via ligation of the T-cell receptor (TCR) initiates important metabolic responses in CD4+ T-cells, whether these metabolic responses are different in CD4+ EM and central memory (CM) subsets compared to naïve (NV) CD4+ T-cells is not well characterised, especially in humans. Human EM T-cells migrate to an area of inflammation and rapidly induce effector function whereas CM T-cells have limited effector function, but can become EM T-cells on secondary stimulation[7]. While there is some evidence that human CD4+ EM T-cells have increased respiratory capacity[8], the majority of the metabolic studies to date have focused on CD8+ memory T-cell subsets[6,9].

It has been shown that CD4+ T-cells undergo metabolic reprogramming by translational remodelling upon long-term (48 h) activation[10,11]. Whilst understanding how T-cells adapt during long-term activation is important, we also need to understand how T-cell metabolism is governed early after activation and its role in predisposing a successful immune response. Currently, the early metabolic response, the key signalling nodes that regulate it and the contribution it makes to T-cell function in activated human CD4+ T-cell subsets has not been characterised.

Studies of human CD8+ T-cell metabolism show that the transition from quiescent to activated T-cell is accompanied by immediate increases in glycolysis in both NV and EM CD8+ T-cells, though the increase in EM CD8+ T-cells is significantly greater than that in NV CD8+ T-cells[6,9]. Whilst glucose is often considered to be the primary fuel metabolised by T-cells upon activation, other fuel sources, most notably glutamine, are essential (e.g. to fuel the TCA cycle) to maintain energy homeostasis and support cellular biosynthetic pathways[12–14]. Indeed, glutaminase, an enzyme involved in glutaminolysis, has been shown to promote murine T-cell differentiation towards the Th17 lineage whilst preventing Th1 development[15].

Herein, we characterise the metabolic phenotypes of human CD4+ NV, EM and CM T-cells. Importantly, we uncover metabolic differences that are not homologous to human CD8+ T-cell subsets[6]. We show that NV CD4+ T-cells differ from their CD8+ counterparts by rapidly engaging glycolysis (supported by Akt activity) and oxygen consumption upon stimulation. We find that TCR-induced NV CD4+ T-cells are heavily reliant on glutaminolysis to drive oxidative phosphorylation (OXPHOS) and this is dependent on signal transducer and activator of transcription 5 (STAT5). Importantly, we define STAT5 as a central node in NV CD4+ T-cell metabolism, disruption of this pathway and the metabolic program it supports impairs T-cell function upon activation, demonstrating the fundamental importance of this early metabolic response.

## Results

**EM and CM T-cells have heightened quiescent metabolism.** Quiescence is an important part of the T-cell life cycle and must be tightly regulated to prevent autoimmune disorders. Therefore, understanding how metabolism is regulated and controlled at rest in CD4+ T-cells, especially the key subsets determined by antigen experience, is important.

To determine the glycolytic activity of the CD4+ subsets in quiescence, extracellular acidification rate (ECAR) was measured using a Seahorse extracellular flux analyzer (Fig. 1a and Supplementary Fig. 1a). There were no significant differences between the resting (basal) ECAR of NV, EM and CM T-cells (Fig. 1b); however, both memory populations had significantly higher maximal ECAR in comparison to NV T-cells, indicating greater glycolytic capacity (Fig. 1c).

To better understand why resting NV T-cells have reduced glycolytic capacity in comparison to EM and CM T-cells, the expression of key glycolysis enzymes were measured using immunoblotting (Fig. 1d). Levels of GLUT1 (the main T-cell glucose transporter[16]), hexokinase II (HKII), phosphofructokinase (PFKP) and lactate dehydrogenase (LDHA) were increased in both EM and CM populations compared to NV T-cells. Of note is the elevated expression of the M2 isoform of pyruvate kinase (PKM2) in EM and CM cells in comparison to NV cells. PKM2 has reduced catalytic activity in comparison to the M1 isoform, preferential expression of PKM2 slows down the conversion of phosphoenolpyruvate to pyruvate, thus allowing the accumulation of upstream metabolites and reduced entry of carbon into the TCA cycle[17]. These data demonstrate that CM and EM CD4+ T-cell subsets have an enzymatic expression profile that supports their glycolytic phenotype in comparison to NV T-cells.

Alongside the glycolytic parameters (Fig. 1a), we measured oxygen consumption rate (OCR) in all three subsets (Supplementary Fig. 1b; Fig. 2a). The memory populations—EM and CM—displayed greater OXPHOS than NV T-cells with higher levels of basal respiration and ATP-linked respiration (Fig. 2b, c). CM T-cells had higher maximal respiration in comparison with NV (Fig. 2d). There were no significant differences in spare respiratory capacity, proton leak or nonmitochondrial respiration between the three populations (Supplementary Fig. 2a–c). While we demonstrate that EM and CM clearly differ metabolically to NV T-cells, no subset preferentially utilised one pathway as the OCR/ECAR ratio calculated using basal respiration and basal ECAR did not differ between them (Fig. 2e).

To better understand why OXPHOS parameters were higher in EM and CM CD4+ T-cells, the mitochondrial content of all subsets was analysed by flow cytometry and confocal microscopy using the specific mitochondrial probe, MitoTracker Green. Total CD4+ T-cells were isolated and incubated with MitoTracker Green as well as anti-CD45RA and anti-CD197 to enable identification of the subsets by flow cytometry. This enables comparison of mitochondrial content of the three subsets within the same donor. EM and CM T-cells had increased mitochondrial content compared to NV T-cells (Fig. 2f). Cell size in relation to mitochondrial content was also measured by confocal microscopy. Isolated subsets—NV, EM and CM from different donors—were incubated with MitoTracker Green, Cell Mask Orange (cell membrane) and DRAQ5 (nucleus) and analysed using confocal microscopy (Fig. 2g). A signal (MitoTracker) to cell area ratio was calculated revealing both memory populations to have a higher ratio in comparison to NV T-cells (Fig. 2h). Increased mitochondrial mass could explain the heightened OXPHOS in the EM and CM compared to the NV subsets at rest.

A recent study by Bantung et al. demonstrated that EM CD8+ T-cells oxidise glucose-derived pyruvate in the mitochondria to sustain elevated rates of OXPHOS, through a mechanism that involves the translocation of hexokinase I to the mitochondrial membrane[9]. Our data suggest that CD4+ EM and CM cells have an increased reliance upon HKII rather than HKI (Fig. 1d). Here, we have used stable isotope tracer analysis (SITA) to determine whether the increased OXPHOS observed in CM and EM CD4+ T-cells is due to an increase in oxidation of glucose-derived

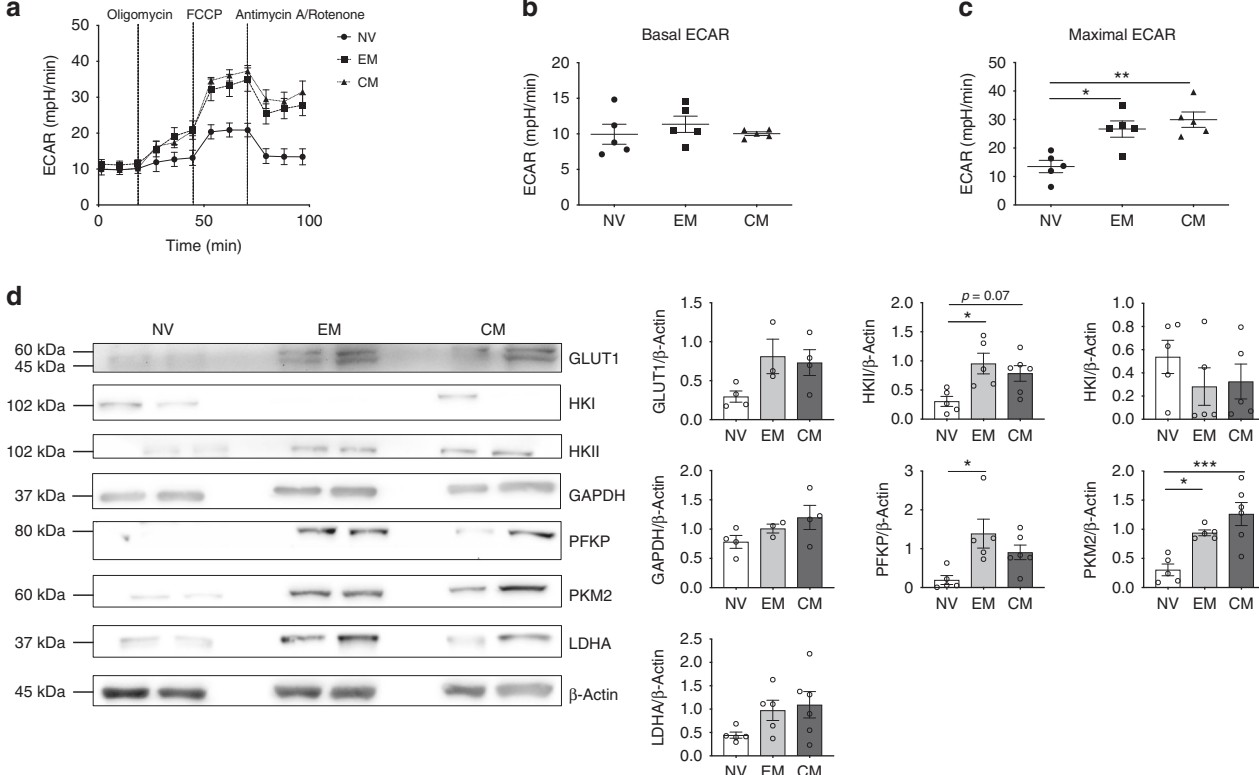

**Fig. 1** Quiescent EM and CM T-cells are metabolically active. **a** Glycolytic stress profile of NV, EM and CM T-cells by measuring ECAR before and following injections of oligomycin (0.75 μM), FCCP (1 μM) and antimycin A and rotenone (1 μM) at the time points indicated. Basal (**b**) and maximal ECAR (**c**) in NV, EM and CM T-cells. **d** Representative immunoblots from two different donors per cell type for GLUT1, HKI HKII, GAPDH, PFKP, PKM2 and LDHA and β-actin. Respective densitometry normalised to β-actin is shown. Data are either representative of five independent experiments (**a-c**) or 3−4 experiments (**d**). Statistical analysis was performed using a non-matching one-way ANOVA with Tukey's multiple comparison test (**b-d**). For non-parametric data, a Kruskal−Wallis test with Dunn's multiple comparisons test was used. Data expressed as mean ± SEM; *$p \leq 0.05$, **$p \leq 0.01$, ***$p \leq 0.001$

pyruvate. In order to determine the contribution of glucose to TCA cycle metabolism in CD4+ T-cell subsets upon activation, NV, EM and CM T-cell populations were activated in the presence of uniformly labelled $^{13}C_6$-glucose for 0.5 or 4 h. Due to cell number restrictions we could only obtain a 4 h time point for CM T-cells. SITA was performed to determine the incorporation of $^{13}C$ from glucose into TCA intermediates and non-essential amino acids (Fig. 2i, j). All subsets integrated glucose-derived $^{13}C$ into TCA intermediates and amino acids after 4 h with no observable difference between the subsets (Fig. 2i, j). Of note, the relative abundance of unlabelled $^{12}C$ intermediates of citrate, α-ketoglutarate and glutamate substantially increased between 0.5 and 4 h for NV CD4+ T-cells, suggesting that other fuels (such as glutamine) are contributing to these metabolite pools (Fig. 2i, j). Therefore, these data show that EM and CM CD4+ T-cells do not have increased oxidation of pyruvate in the TCA cycle relative to NV cells, arguing that EM and CM CD4+ T-cells sustain elevated OXPHOS using fuels other than glucose.

**CD4+ T-cells rapidly engage glycolysis upon activation.** Having established that EM and CM CD4+ T-cells have elevated glycolytic and oxidative capacity at rest compared to NV CD4+ T-cells, we proceeded to determine how these cells responded metabolically to activation. Following activation with anti-CD3/CD28 stimulation, extracellular flux analysis was used to monitor changes in ECAR and OCR over 5 h. A final injection of 2-deoxy-D-glucose (2-DG) arrested glycolysis. All three subsets demonstrated an early increase in glycolytic rate that was sustained over 5 h (Fig. 3a). The differences in ECAR and OCR immediately following stimulation (fold change) and following 257 min of

stimulation (pre/post) were analysed; the parameters used for these calculations are indicated in Supplementary Fig. 3a.

Human EM CD8+ T-cells have a greater immediate glycolytic response in comparison to NV T-cells[6], but this is not the case in CD4+ memory T-cells. In fact, the increase in glycolysis and oxygen consumption rate was greatest in the NV T-cells, which was significantly greater than in EM CD4+ T-cells (Fig. 3a-b, Supplementary Fig. 3a, b). ECAR was increased soon after activation for all subsets (Fig. 3c). As the assay progressed, ECAR steadily increased for both memory populations. Despite these apparent kinetic differences, all three subsets increase their glycolytic rate from baseline post activation (Fig. 3d–f). Oxygen consumption was also increased upon activation in NV T-cells, but not in EM and CM T-cells (Fig. 3g, Supplementary Fig. 3c). In addition, NV T-cells maintained their OCR, whereas both EM and CM T-cells reduced oxygen consumption for the duration of the assay (Fig. 3h–j). We also discovered that the early increase in ECAR in NV cells is dependent on CD28 (the combined effect of CD28 and CD3 remains greater); however, the early OCR switch was entirely dependent on both CD3 and CD28 stimulation (Supplementary Fig. 3d, e). Therefore, while all CD4+ T-cell subsets underwent a shift towards an increased glycolytic rate following TCR activation, the kinetics were different for NV versus memory T-cells. NV CD4+ T cells engage a rapid switch to a more glycolytic metabolic phenotype, whereas EM and CM T-cells engage a more gradual glycolytic response (Fig. 3k). Together, these data demonstrate that NV CD4+ T-cells engage metabolic adaptation following TCR stimulation involving both increased glycolysis and OXPHOS that is not observed in the memory T-cell subsets.

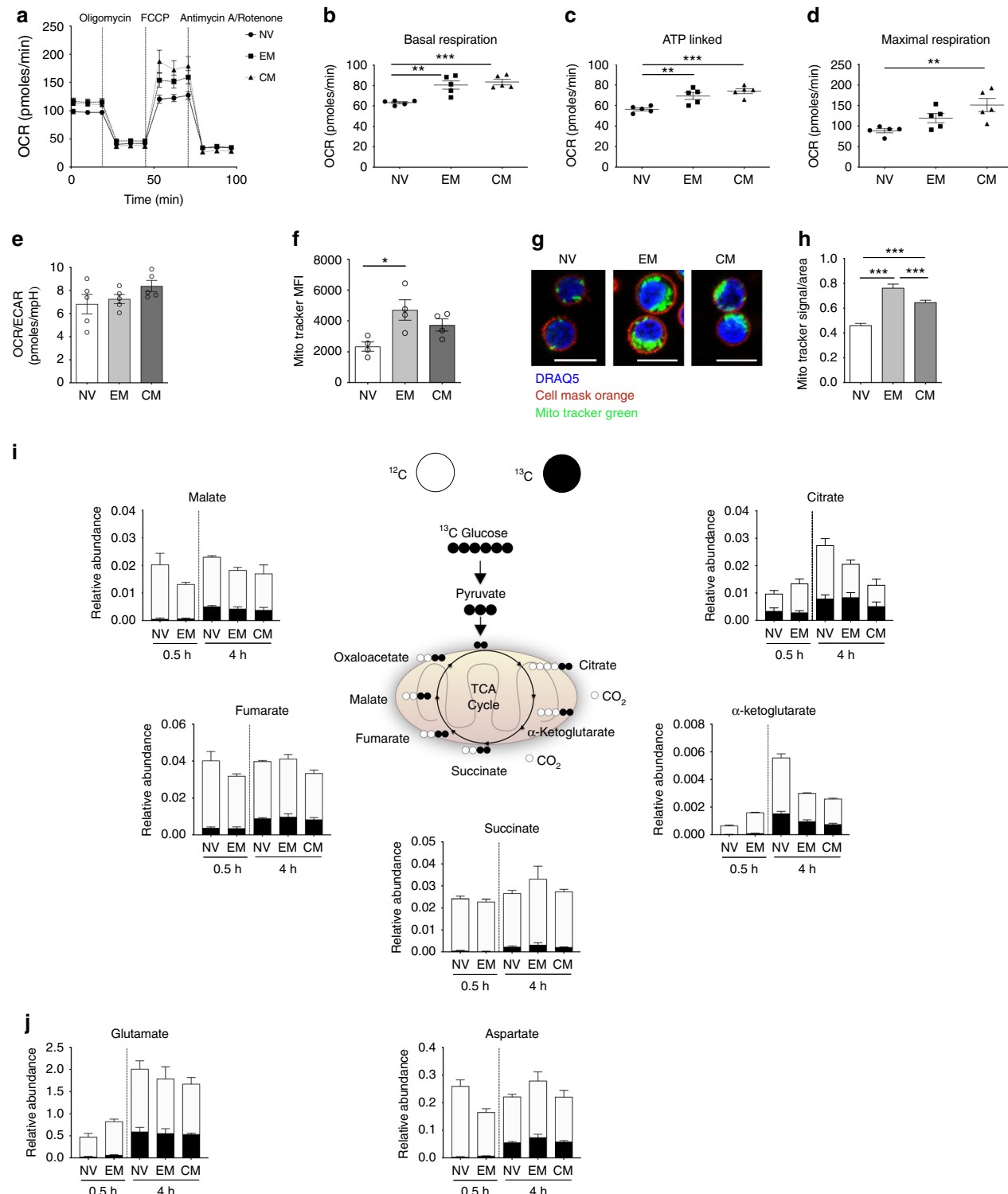

**Fig. 2** Activated CD4+ T-cell subsets incorporate glucose into the TCA cycle. **a** Oxidative phosphorylation profile of NV, EM and CM T- cells by measuring OCR before and following injections of oligomycin (0.75 μM), FCCP (1 μM) and antimycin A and rotenone (1 μM). Basal respiration (**b**), ATP-linked respiration (**c**), maximal respiration (**d**) and OCR/ECAR ratio (**e**) in NV, EM and CM T- cells. **f** Mitochondrial content of NV, EM and CM CD4+ T-cells measured using MitoTracker. **g** Representative images of NV, EM and CM T-cells stained with DRAQ5 (nucleus), cell mask orange (plasma membrane) and MitoTracker green (mitochondria) scale bar = 10 μm and **h** corresponding MitoTracker signal to area ratios. **i** Uniformly labelled $^{13}$C-glucose incorporation into T-cell metabolites via the TCA cycle in NV, EM and CM T-cells activated for 0.5 and 4 h. Relative abundance of $^{12}$C and $^{13}$C including citrate, α-ketoglutarate, succinate, fumarate and malate. **j** Relative abundance of $^{12}$C and $^{13}$C in non-essential amino acids glutamate and aspartate . Data are representative of either five independent experiments (**a**–**e**), four independent experiments (**f**), three experiments with <100 cells analysed as technical replicates (**h**) or three independent experiments (**i**, **j**). Statistical analysis was performed using a nonmatching one-way ANOVA with Tukey's multiple comparison test (**b**–**h**). For non-parametric data, a Kruskal−Wallis test with Dunn's multiple comparisons test was used. Data expressed as mean ± SEM; *$p \leq 0.05$, **$p \leq 0.01$, ***$p \leq 0.001$

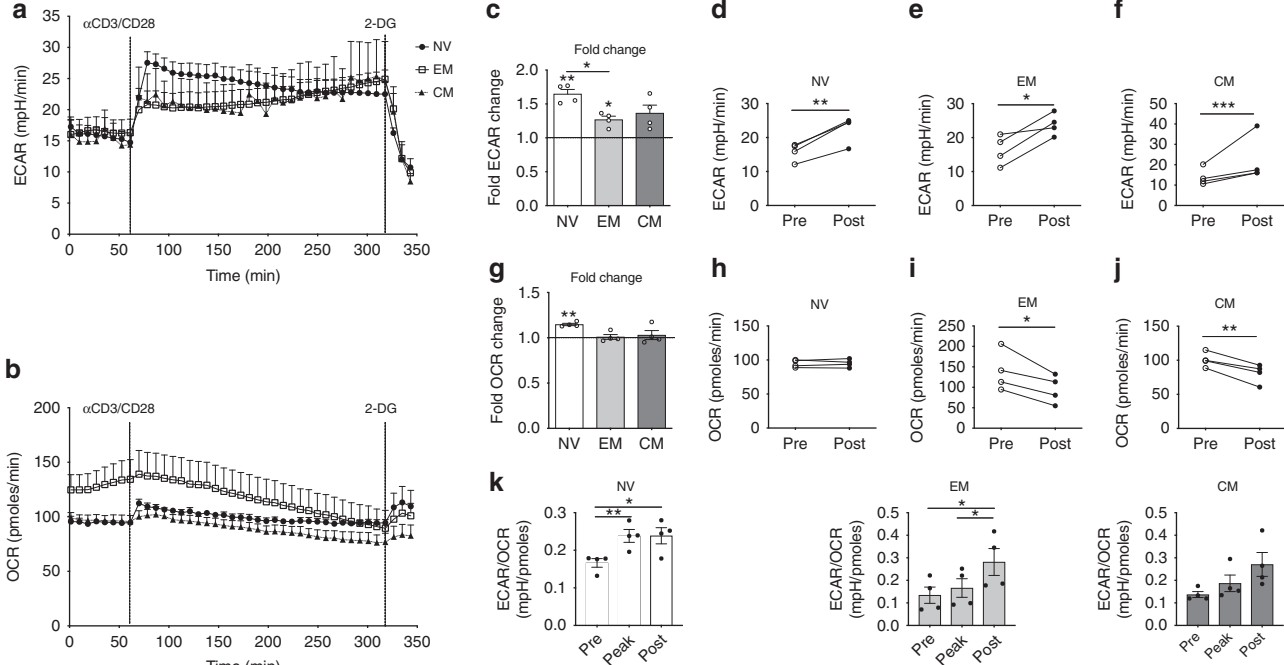

**Fig. 3** NV T-cells have a greater metabolic response upon activation. **a** ECAR and **b** OCR in NV, EM and CM T- cells upon stimulation with anti-CD3/CD28. **c** Fold ECAR change in NV, EM and CM T-cells upon activation. **d** NV, **e** EM and **f** CM ECAR 'Pre' and 'Post' activation taken from the measurements in the indicated boxes in Supplementary Fig. 3a. **g** Fold change in OCR in NV, EM and CM T-cells upon activation. **h** NV, **i** EM and **j** CM OCR 'Pre' and 'Post' activation taken from the measurements in the indicated boxes in Supplementary Fig. 3a. **k** ECAR/OCR ratio of 'Pre', 'Peak' and 'Post' activation kinetics of NV, EM and CM T-cells. Statistical analysis was performed using a non-matching one-way ANOVA with Tukey's multiple comparison test (**c**, **g**, **k**). For non-parametric data, a Kruskal−Wallis test with Dunn's multiple comparisons test was used. A one-sample t-test was also used comparing values to a theoretical value of 1 (**c**, **g**). A paired t-test was also used (**d–f**, **h–j**) whereby a Wilcoxon matched-pairs signed rank test was applied to non-parametric data. Data are representative of four experiments (**a–k**) and are expressed as mean ± SEM; *$p \leq 0.05$, **$p \leq 0.01$, ***$p \leq 0.001$

**TCR-induced glycolysis is dependent on Akt in NV T-cells**. We next investigated the mechanisms involved in mediating the pronounced metabolic responses in TCR-activated NV T-cells. TCR-induced PI3K-Akt signalling is important in both CD4+ and CD8+ T-cells and has been linked to the robust immediate glycolytic response in CD8+ EM T-cells[6,18,19]. TCR stimulation induced the phosphorylation of Akt on threonine 308 and serine 473, with more robust activation observed in NV T-cells (Fig. 4a–c). Robust activation is also apparent at earlier time points (Supplementary Fig. 4a). To investigate whether Akt was required for the glycolytic response in NV CD4+ T-cells the allosteric inhibitor, Akti-1/2, was used to inhibit activity prior to TCR triggering (efficacy of the inhibitor at the concentration used is demonstrated in Supplementary Fig. 4b). Akti-1/2 incubation prevented the increase in ECAR following TCR stimulation in NV CD4+ T-cells (Fig. 4d). Akti-1/2 had no impact upon TCR-induced OCR in these cells (Fig. 4e). Therefore, we next considered the TCR-induced signalling that might underpin the early increase in OCR in NV CD4+ T-cells.

**Lck-dependent STAT5 is required for TCR-induced OXPHOS.** We observed that TCR stimulation induced the phosphorylation of STAT5 on tyrosine 694 in NV, CM and EM CD4+ T-cells (Fig. 5a). Robust activation is also apparent at earlier time points (Supplementary Fig. 4a). As STAT5 phosphorylation upon TCR ligation has not been widely reported, we confirmed the specificity of the STAT5 antibody using IL-2 and IL-7 (known inducers of STAT5 phosphorylation) as positive controls (Supplementary Fig. 5)[20]. STAT5 phosphorylation was dependent on TCR-induced lck activity and was abolished by the lck inhibitor 4-

Amino-5-(4-phenoxyphenyl)-7H-pyrrolo[3,2-d]pyrimidin-7-yl-cyclopentane (Fig. 5b).

As there is an emerging role for STAT proteins in mitochondrial metabolism[21,22], we next considered whether TCR-induced STAT5 is required for the early increase in OXPHOS observed in NV CD4+ T-cells. A commonly used STAT5 inhibitor N′-((4-Oxo-4H-chromen-3-yl)methylene)nico-tinohydrazide (STAT5i) was optimised (Supplementary Figs. 5b and 6) and used (Fig. 5c–h) to investigate the role of STAT5 in TCR-induced cellular metabolism[23]. ECAR and OCR were measured pre- and post activation with anti-CD3/CD28 in the presence of STAT5i or a vehicle control (Fig. 5c–h). Inhibition of STAT5 reduced the TCR-induced increase in OCR in NV T-cells (Fig. 5c), but had no impact on OCR in CM or EM CD4+ T-cells (Fig. 5d, e). Inhibition of STAT5 also reduced the TCR-induced increase in ECAR in NV CD4+ T-cells but not to the same extent as observed upon Akt inhibition (Figs. 4d, 5f, i). In contrast, inhibition of STAT5 did not affect ECAR in TCR-stimulated CM and EM T-cells (Fig. 5g–i). Consistent with these data, STAT5 inhibition resulted in a substantial decrease in both glucose uptake and extracellular lactate production in TCR-stimulated NV CD4+ T-cells (Fig. 5j). Next we investigated whether IL-2 and IL-7 (known inducers of Akt and STAT5[18,24]) would individually induce an increase in OCR or ECAR (Fig. 5k–l). However, unlike TCR stimulation, exposure to IL-2 or IL-7 did not affect the metabolic rates of NV T-cells. In order to determine whether the STAT5 phosphorylation is directly downstream of TCR activation or activated by autocrine IL-2 secretion, we cultured NV T-cells with a common γ chain antibody or respective isotype control for 0.5 and 3 h. We found there was no significant decrease in STAT5 phosphorylation between the

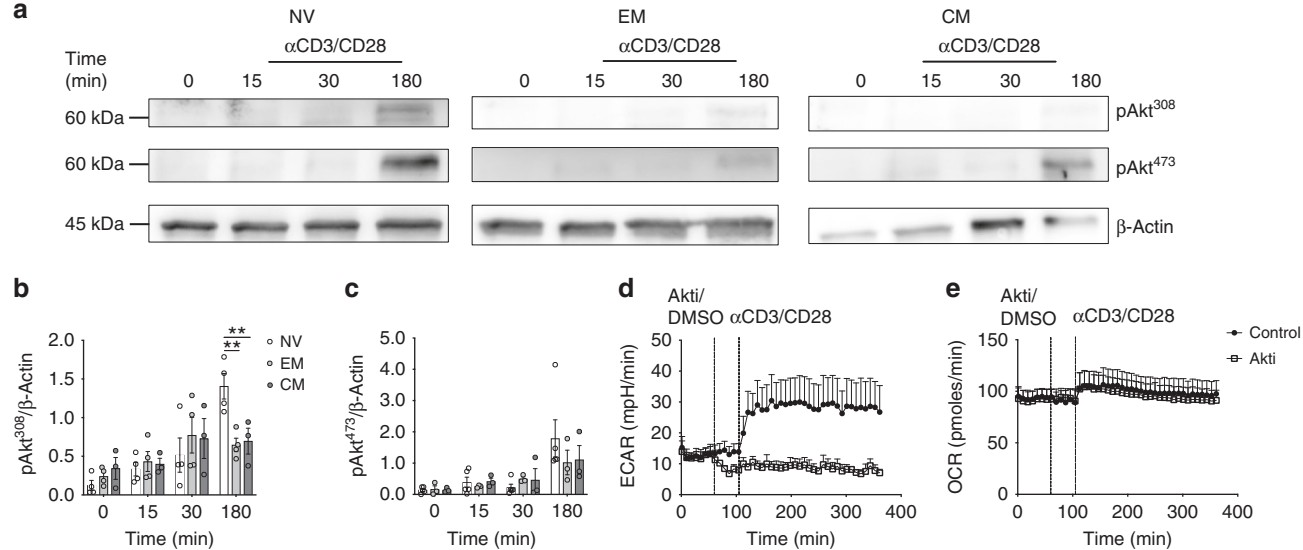

**Fig. 4** Early T-cell activation requires Akt phosphorylation. **a** Immunoblot for pAkt Thr308 and Ser473 and β-actin in NV, EM and CM T-cells following 0, 15, 30 and 180 min of activation with anti-CD3/CD28. Densitometry of pAkt Thr308 (**b**) and pAkt Ser473 (**c**) for NV, EM and CM T-cells normalised to β-actin. ECAR (**d**) and OCR (**e**) in NV CD4+ T-cell upon treatment with either 10 μM Akt1/2 kinase inhibitor or vehicle control (DMSO) and anti-CD3/CD28 at the times indicated. Data are representative (**a–c**) of 3–5 experiments with one representative immunoblot sample of 3–5 is shown, (**d**, **e**) three experiments. Statistical analysis was performed using a nonmatching two-way ANOVA with Sidak's multiple comparison test (**b**, **c**). Data are expressed as mean ± SEM; **$p \leq 0.01$

control or γ chain antibody-treated NV T-cells, indicating that the TCR directly activates STAT5 (Fig. 5m, n and Supplementary Fig. 7).

**STAT5 regulates glutaminolysis in NV CD4+ T-cells**. Next we focused on the role of STAT5 in the early metabolic response in TCR-triggered NV CD4+ T-cells. Glutamine has been described as an important fuel for sustaining elevated rates of OXPHOS in activated murine T-cells[14]. Therefore, we investigated the role of STAT5 in glutamine metabolism in NV CD4+ using the STAT5 inhibitor. NV T-cells were activated with anti-CD3/CD28 in media containing uniformly labelled $^{13}$C-glutamine ($^{13}C_5$-glutamine) in the presence of STAT5i (Fig. 6a) before extraction of cellular metabolites for mass spectrometry analysis.

Our data show incorporation of $^{13}$C into all TCA intermediates, demonstrating that glutamine is important in fuelling the TCA cycle in TCR-activated NV CD4+ T-cells (Fig. 6a, b). Incubation with STAT5i resulted in a decrease in the incorporation of $^{13}$C from glutamine into TCA cycle intermediates (Fig. 6b), consistent with the observed decrease in OCR (Fig. 5c). In particular there was a striking decrease in $^{13}$C incorporation into α-ketoglutarate, the point at which glutamine carbon enters the TCA cycle. The levels of $^{13}$C in intracellular glutamine and glutamate were also substantially decreased arguing that STAT5i inhibits glutaminolysis in TCR-stimulated NV T-cells (Fig. 6c). Interestingly, these data show that in TCR-stimulated NV CD4+ T-cells there was a greater incorporation of $^{13}$C from glutamine into α-ketoglutarate and citrate over the other TCA intermediates, suggesting that TCR-activated NV T-cells are also metabolising glutamine by reductive decarboxylation, converting α-ketoglutarate directly to citrate (Fig. 6a, b). To provide further insight into the mechanism of glutamine incorporation into the TCA cycle, the mass isotopologue distributions of TCA intermediates were analysed. The metabolite pools of succinate, fumarate, malate and citrate were all composed of a substantial proportion of the $m + 4$ mass isotopologue indicative of conventional glutaminolysis. The distribution of the $m + 4$ mass isotopologue was reduced in all cases in the presence of STAT5i

(Fig. 6d). The presence of the $m + 5$ mass isotopologue (14%) in the metabolite pool of citrate indicates that reductive decarboxylation of α-ketoglutarate to citrate is also taking place. The abundance of the $m + 5$ mass isotopologue (indicative of reductive decarboxylation) was abolished by STAT5 inhibition.

Glutamine is not only used to fuel ATP production via the TCA cycle but can also be used to support biosynthesis. Citrate generated from glutamine through the TCA cycle and directly via reductive carboxylation can be exported from the mitochondria to fuel biosynthetic processes, such as providing acetyl-CoA for fatty acid synthesis and amino acids for protein synthesis. For example, glutamine is used to generate the amino acid aspartate which is essential for nucleotide biosynthesis. In addition, glutamine can be used for the production of other amino acids such as proline and this is independent of the TCA cycle. Indeed, our data show $^{13}$C-glutamine was incorporated into the aspartate and proline metabolite pools in activated NV T-cells and levels of these glutamine-derived amino acids were substantially decreased when STAT5 was inhibited (Fig. 6e). Together, these data argue that glutamine is an important fuel for supporting ATP production and biosynthesis in TCR-stimulated NV T-cells and that STAT5 activity is required.

Next we considered the molecular mechanisms linking STAT5 to the regulation of glutamine metabolism in TCR-stimulated NV CD4+ T-cells. Given that there is crosstalk between the STAT5 and mTORC1 pathways and mTORC1 is a known regulator of glutaminolysis[25,26], we investigated whether STAT5 inhibition might affect mTORC1 signalling. Indeed, NV T-cells treated with STAT5i have reduced phosphorylation of the mTORC1 target p70S6K and the downstream p70S6K substrate, S6 ribosomal protein in comparison to the vehicle controls (Fig. 6f).

There are a number of examples where lymphocytes demonstrate metabolic plasticity and adjust their metabolic pathways as needed to maintain energy homoeostasis[12]. Therefore, we considered whether treating TCR-activated NV T-cells with STAT5i might change the way that they metabolise glucose when glutamine metabolism is disrupted. Indeed, upon incubation with STAT5i there was a substantial increase in the

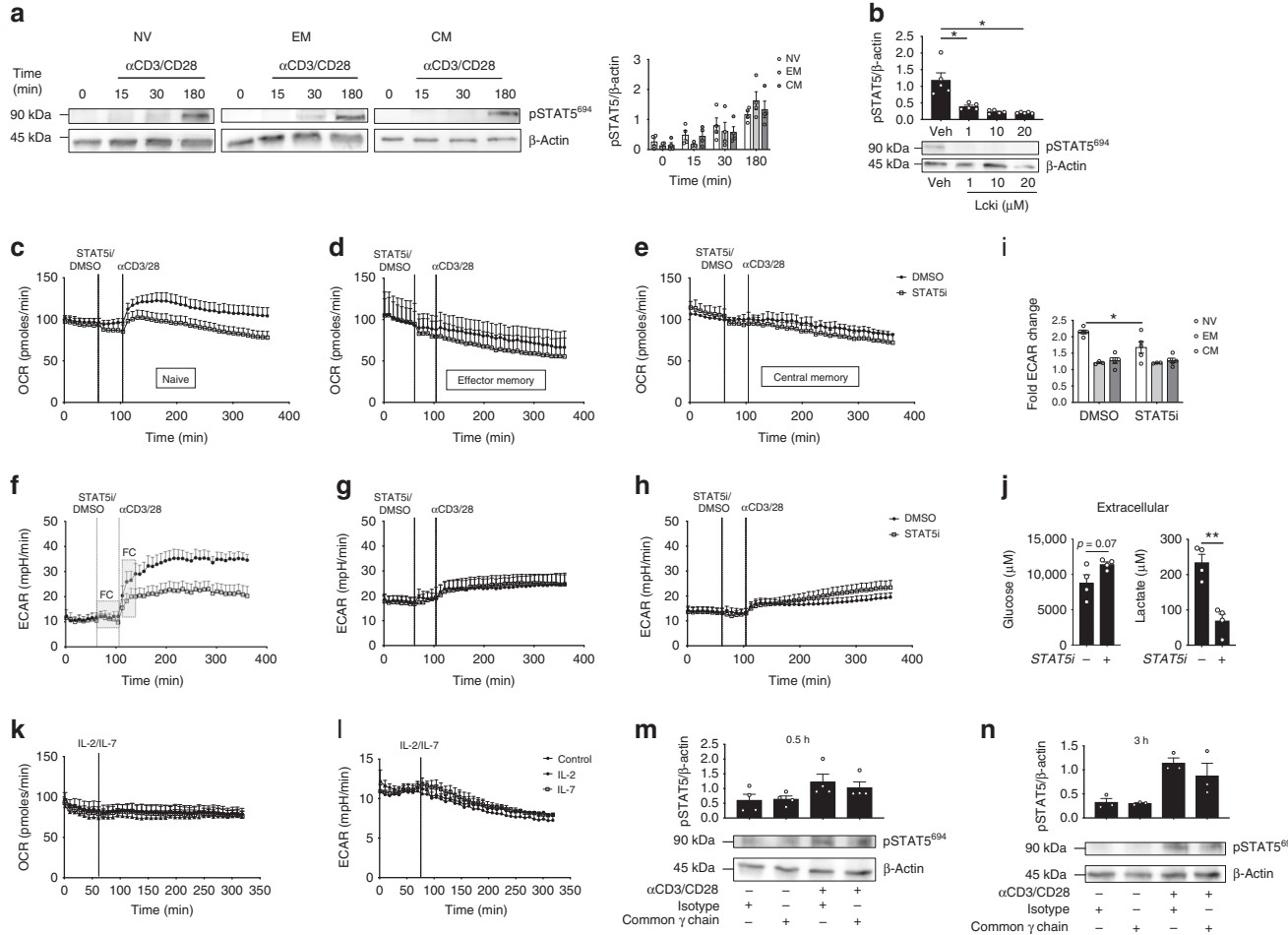

**Fig. 5** STAT5 orchestrates the metabolic switch in activated NV T-cells. **a** Immunoblot for pSTAT5 Tyr694 and β-actin in NV, EM and CM T-cells following 0, 15, 30 and 180 min of activation with anti-CD3/CD28. Densitometry of pSTAT5 Tyr694. **b** Immunoblot for pSTAT5 Tyr694 and β-actin in NV T-cells activated with anti-CD3/CD28 in the presence (1, 10 and 20 μM) or absence (Veh) of a lck inhibitor. OCR (**c–e**) and ECAR (**f–h**) was measured in NV (**c**, **f**), EM (**d**, **g**) and CM (**e**, **h**) CD4+ T-cells in the absence (DMSO) or presence of a STAT5 inhibitor (100 μM) before cells were activated with anti-CD3/CD28 at the indicated time points. **i** Fold change in ECAR upon activation calculated using the measures in the FC box. **j** Extracellular glucose and lactate production in activated NV T-cells (anti-CD3/CD28) in the absence or presence of STAT5i (100 μM) for 4 h. OCR (**k**) and ECAR (**l**) of NV T-cells activated with either IL-2 or IL-7 (10 ng/mL). Immunoblot of pSTAT5 Tyr694 and β-actin in NV T-cells activated with anti-CD3 (2 μg/mL) and anti-CD28 (20 μg/mL) with common γ chain antibody or isotype control (1 μg/mL) for **m** 0.5 or **n** 3 h. Statistical analysis was performed using a non-matching two-way ANOVA with Sidak's multiple comparison test (**a**, **i**), a paired $t$- test (**j**) or a matched Friedman test with Dunn's multiple comparisons test (**m**, **n**). Data are representative of **a** 3–5 experiments with one representative immunoblot sample of 3–5 is shown, five (**b**, **c**, **e**, **f**, **h**), three (**d**, **g**, **n**), four (**j**, **m**) or two independent experiments (**k**, **l**) and expressed as mean ± SEM; *$p \leq 0.05$, **$p \leq 0.01$

incorporation of [13]C-glucose into TCA intermediates (Fig. 7a, b), but not amino acids (Fig. 7c). It has recently been shown that glucose can fuel OXPHOS in activated lymphocytes via the citrate-malate shuttle (indicated by the predominance of $m + 2$ citrate)[27]; however, we do not observe this here (Fig. 7b). Despite the compensation we observe here that glucose alone is not sufficient to maintain flux through the TCA cycle as it must be replenished by anaplerosis, most commonly from glutaminolysis. Indeed, STAT5i-treated NV T-cells have reduced levels of OXPHOS (Figs. 5a, 7a).

**STAT5-mediated glutaminolysis supports NV T-cell function.** Thus far we have shown that STAT5-dependent glutamine metabolism is important for both energy production and biosynthesis in TCR-activated NV CD4+ T-cells. To determine the role of the STAT5-dependent early metabolic program on CD4+ T-cell function, we impaired oxidative metabolism (either by depriving the cells of glutamine or by using metabolic inhibitors) and measured IL-2 production. IL-2 production was effectively abolished in

NV CD4+ T-cells upon incubation with STAT5i or in the absence of glutamine (Fig. 8a, b, Supplementary Fig. 8a, b). We next determined the relative importance of the different glutamine-fuelled metabolic pathways for TCR-activated IL-2 production (Fig. 8c). The importance of generating α-ketoglutarate from glutamine (for TCA cycle anaplerosis) to support IL-2 production was confirmed using an inhibitor that prevents the conversion of glutamine to glutamate, 6-diazo-5-oxo-L-norleucine (DON), and an inhibitor preventing the conversion of glutamate to α-ketoglutarate, aminooxyacetic acid (AOA). Both inhibitors potently blocked TCR-induced IL-2 production from NV CD4+ T-cells (Fig. 8d, e, Supplementary Fig. 8c, d). These data argue that α-ketoglutarate production is crucial for TCR-induced T-cell function. We determined that α-ketoglutarate is sufficient to support T-cell function in the absence of glutamine by rescuing IL-2 production with a membrane permeable α-ketoglutarate (dimethyl 2-oxoglutarate - DMK) (Fig. 8g, Supplementary Fig. 8e). We next addressed whether α-ketoglutarate supports IL-2 production in NV T-cells by supporting energy production or supporting cellular biosynthesis. While

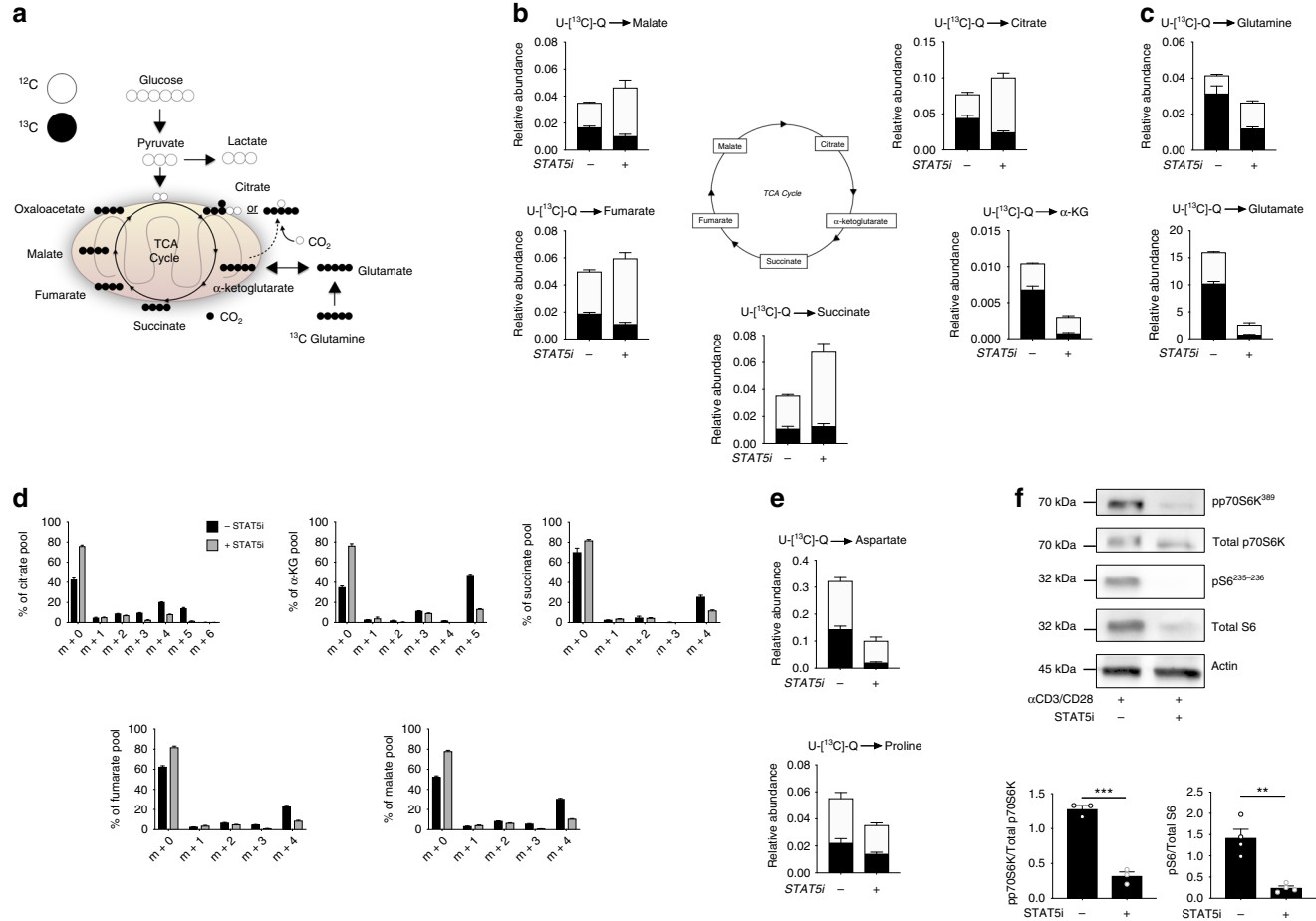

**Fig. 6** STAT5 regulates glutaminolysis in an mTORC1-dependent manner. **a** Schematic summarising incorporation of uniformly labelled $^{13}$C-glutamine into the TCA cycle. NV T-cells were activated (anti-CD3/CD28) in the presence or absence of STAT5i (100 μM) for 4 h. **b** Relative abundance of glutamine-derived $^{12}$C and $^{13}$C TCA cycle metabolites, citrate, α-ketoglutarate, succinate, fumarate and malate. **c** Relative abundance of glutamine-derived $^{12}$C and $^{13}$C glutamine and glutamate. **d** Mass isotopologue distributions (MID) of TCA cycle intermediates, citrate, α-ketoglutarate, succinate, fumarate and malate. **e** Relative abundance of glutamine-derived $^{12}$C and $^{13}$C amino acids aspartate and proline. **f** Immunoblot for p70S6K and ribosomal S6 phosphorylation (pS6) in NV T-cells following 4 h activation with (anti-CD3/CD28) in the absence and presence of STAT5i (100 μM). β-actin was used as a loading control. Statistical analysis was performed using a paired *t*-test (**f**). Data are representative of four experiments (**b**–**e**) or three experiments (**c**) and are expressed as mean + SEM. **$p \leq 0.01$, ***$p \leq 0.001$.

inhibition of ATP synthesis (by low dose oligomycin) significantly reduced IL-2 production, an inhibitor of ATP citrate lyase (ACLY) had no effect in TCR-stimulated NV CD4+ T-cells (Fig. 8g, h, Supplementary Fig. 8f, g). Taken together, these results suggest that STAT5-dependent glutamine metabolism supports NV CD4+ T-cell function by supporting ATP production (Fig. 8i).

### Discussion
In this study we characterise the metabolic phenotypes of NV, EM and CM CD4+ T-cells and investigate the mechanisms underlying these processes. Importantly, we elucidate a novel role for STAT5 in the control of glutaminolysis and TCA cycle metabolism in human T-cells. We also establish how this metabolic program promotes and supports NV T-cell function.

We find that memory CD4+ T-cells have elevated activity of metabolic pathways compared to NV CD4+ T-cells in quiescence, in terms of their capacity to engage glycolysis and mitochondrial respiration. These data are the first to demonstrate increased metabolic processes in CM CD4+ T-cells and are in agreement with another recent study with respect to EM cells[8].

The metabolic phenotype of CD4+ memory cells is distinct to that described for CD8+ EM memory T-cells. CD4+ CM and

EM T-cells have elevated levels of basal OXPHOS and an increased capacity for glycolysis that reflects increased mitochondrial mass and increased expression of key glycolytic enzymes. The glycolytic machinery in CD8+ memory T-cells is not increased and while CD8+ EM T-cells have a slightly increased respiratory capacity they do not have elevated levels of basal OXPHOS[6]. The metabolic response of CD4+ versus CD8+ memory T-cells following TCR triggering is also distinct. CD8+ memory T-cells engage an immediate glycolytic response that is mediated by increased GAPDH activity while CD4+ CM and EM T-cells gradually undergo a shift to a glycolytic phenotype, simultaneously increasing glycolysis and decreasing OXPHOS. The different kinetics of the metabolic response to TCR stimulation are likely due to the fact that CD4+ CM and EM T-cells are already in a heightened metabolic state prior to activation and so can meet the metabolic demands without engaging in an early metabolic response. CD8+ EM T-cells oxidise glucose in the mitochondria to sustain elevated OXPHOS[9]. However, elevated OXPHOS in CD4+ EM and CM T-cells is not due to increased glucose-mediated fuelling of the TCA cycle. As such, our study reveals important differences in the metabolic phenotypes of CD8+ and CD4+ memory T-cells.

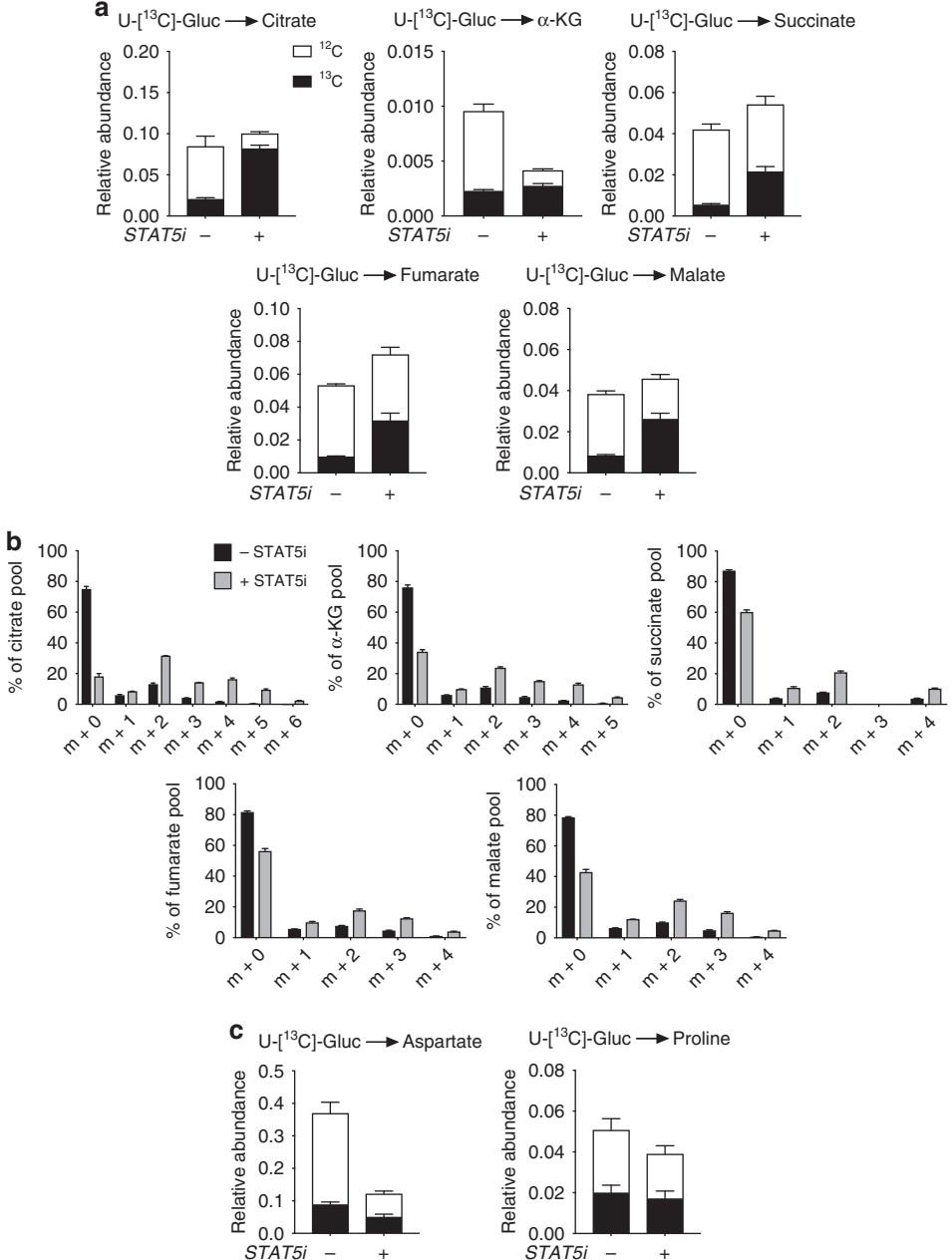

**Fig. 7** Disrupted glutaminolysis causes compensatory TCA cycling. **a** NV T-cells were activated (anti-CD3/CD28) in the presence or absence of STAT5i (100 μM) for 4 h. Relative abundance of glucose-derived $^{12}C$ and $^{13}C$ TCA cycle metabolites citrate, α-ketoglutarate, succinate, fumarate and malate. **b** Mass isotopologue distributions (MID) of TCA cycle intermediates citrate, α-ketoglutarate, succinate, fumarate and malate. **c** Relative abundance of glucose-derived $^{12}C$ and $^{13}C$ amino acids aspartate and proline. Data are representative of four experiments and are expressed as mean + SEM

In contrast to CM and EM CD4+ T-cells, and indeed NV CD8+ T-cells[6], NV CD4+ T-cells engage an early metabolic response following TCR/CD28 stimulation that involves increased rates of glycolysis and OXPHOS. While this rapid glycolytic response following TCR stimulation has been observed in murine and human CD8+ T-cells, this is the first time it has been described in NV CD4+ T-cells. We found that increased glycolysis in activated NV T-cells was dependent on CD28 ligation, whereas in murine CD8+ NV and previously activated T-cells, increased glycolysis was independent of CD28 in the short term[28]. This result is supported by another study that showed an early, although transient, increase in OCR in CD4+ T-cells from patients with systemic lupus erythematosus[29]. However, these data are difficult to interpret as the analysis was performed on total CD4+ T-cells.

We observed an inverse relationship between ECAR and OCR in EM and CM T-cells. Upon activation, EM and CM T-cells gradually increased ECAR and reduced OCR. This phenomenon could reflect the translocation of HKII toward the mitochondria upon stimulation, thus shifting metabolism towards glycolysis; this has been observed in other cell types[30,31]. The initial burst of glycolysis observed in NV T-cells could be explained by elevated levels of mitochondria-bound HKI, which has previously been linked to increased glycolysis in murine memory T-cells[32]. Taken together, our data highlight the metabolic demands placed on NV CD4+ T-cells upon immune activation to facilitate rapid and robust responses.

It is interesting that the kinetics of the glycolytic response in NV CD4+ T-cells are actually very similar to those observed in

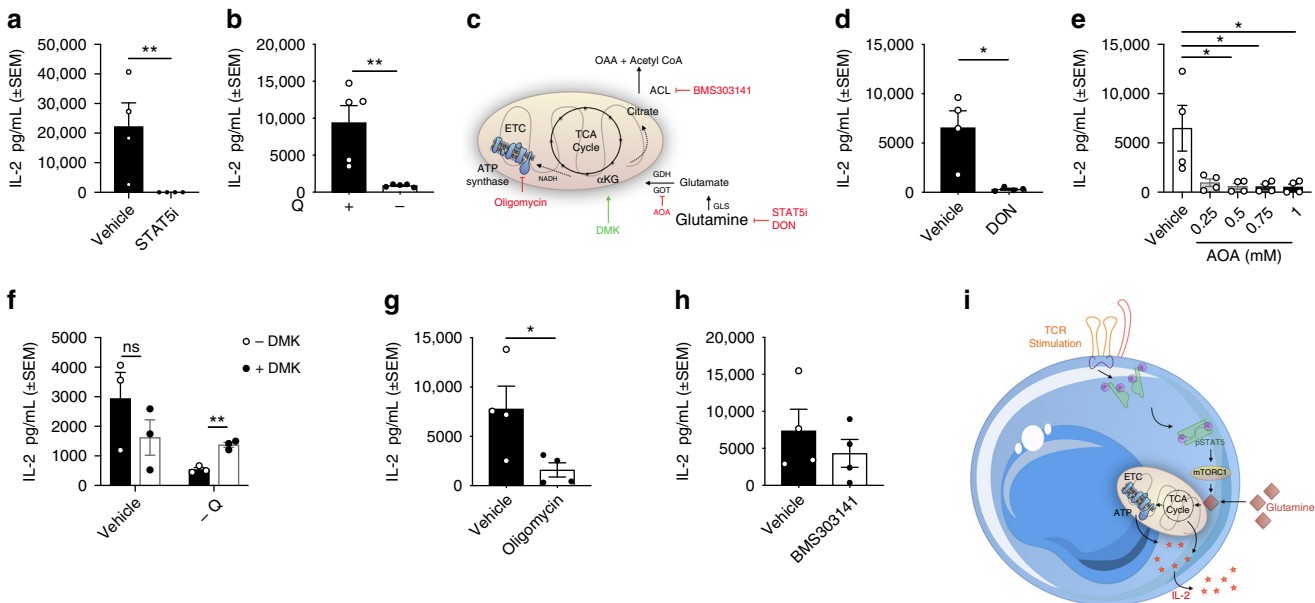

**Fig. 8** Disruption glutaminolysis impacts on NV T-cell IL-2 production. IL-2 production of NV T-cells cultured with anti-CD3 (2 µg/mL) and anti-CD28 (20 µg/mL) in the presence or absence of **a** STAT5 inhibitor (STAT5i; 100 µM), **b** glutamine withdrawal (- Q). **c** Schematic of the different mechanisms used to investigate glutamine metabolism in human NV T-cells. IL-2 production of NV T-cells cultured with **d** DON (50 µM) or **e** aminooxyacetic acid (AOA; 0.25–1 mM). **f** NV T-cells activated and treated as in **b** with the addition of dimethyl 2-oxoglutarate (DMK; 0.3 mM) with downstream analysis of IL-2 production. NV T-cell cultured with **g** oligomycin (100 nM) and **h** BMS303141 (1 µM) with IL-2 production measured. **i** Schematic of NV T-cell activation via the T-cell receptor leading to downstream STAT5 phosphorylation. Glutaminolysis is regulated by STAT5 in an mTORC1-dependent manner. Statistical analysis was performed using an unpaired *t*-test (**a**, **b**, **d**, **f**, **g**), a Kruskal−Wallis test (**e**) or a one-way ANOVA (**g**). Data are representative of four experiments (**a**, **d–e**, **h**), five experiments (**b**) and three experiments (**f**) expressed as mean ± SEM; *$p \leq 0.05$, **$p \leq 0.01$

CD8+ EM T-cells; both cell types respond to TCR activation with an early increase in lactate production[6]. Additionally, the data herein demonstrate that the early glycolytic response in NV CD4+ T-cells is completely dependent on Akt activity as is shown to be the case in CD8+ EM T-cells[6]. This is in contrast to activated murine NV CD8+ T-cells where the early glycolytic switch is mediated by pyruvate dehydrogenase kinase 1 (PDHK1) independent of Akt[28].

A unique feature of NV CD4+ T-cells is the very early increase in OXPHOS that accompanies TCR activation, as this has not been described in other T-cell subsets to date. The fact that this OXPHOS response was not affected by Akt inhibition (in contrast to the increase in glycolysis) suggested that fuels other than glucose must contribute to the observed increase in mitochondrial respiration. Indeed, we show that glutamine is an essential fuel for activated NV CD4+ T-cells and is used to support OXPHOS through TCA cycle anaplerosis and amino acid biosynthesis—including aspartate (for protein and nucleotide synthesis) and proline (for protein synthesis).

Our data demonstrate that 4 h following activation, glutamine is used to support both TCA cycle anaplerosis and citrate biosynthesis via reductive carboxylation in NV CD4+ T-cells. This is consistent with work carried out in T-cells following 48 h activation[10] and complements previous work that demonstrates human T-cell activation is accompanied by an increase in glycolytic machinery[1,33]. Glutamine-dependent reductive carboxylation has been best described in tumour cells where it supports anabolic processes including lipid synthesis[34,35], or maintaining mitochondrial redox homoeostasis[36]. Further work is required to establish exactly what role glutamine-dependent reductive carboxylation plays in TCR-stimulated NV CD4+ T-cells.

While glutaminolysis has been studied in murine T-cells, here we describe glutamine utilisation in human T-cell subsets using a metabolomic approach. This is significant because inhibitors of

glutaminase, the first enzymatic step in glutaminolysis, are in development for the treatment of cancer and hyperglycaemia. As such, it will be important to understand how these drugs could impact patient immunity[37–39].

A key and novel insight of our study is the characterisation of STAT5 as a node linking TCR stimulation to increased OXPHOS (via glutaminolysis) in NV CD4+ T-cells. While STAT5 activation has been classically associated with cytokine signalling and JAK kinases, the phosphorylation of STAT5 downstream of TCR stimulation has also been reported[20]. Here, we show that TCR-induced STAT5 phosphorylation was blocked by inhibition of the TCR-activated lck tyrosine kinase. Overexpression of lck in B-cell leukaemia has been shown previously to drive STAT5 activation[40]. In contrast to TCR stimulation, we did not observe any metabolic changes in NV T-cells stimulated with IL-2 or IL-7; this could reflect the fact that NV T-cells are constantly bathed in cytokines within the lymph nodes, preventing the occurrence of unnecessary activation. Therefore, we can conclude that STAT5 is required but not sufficient for NV T-cell activation.

A role for the STATs in mitochondrial metabolism has begun to emerge[21,41]. Here we report a profound effect of STAT5 inhibition on glutamine metabolism in CD4+ NV T-cells. The most striking defect observed following STAT5i treatment was in the depletion of glutamate, the product of glutaminase and the first step of glutaminolysis. This suggests that STAT5 may regulate glutamine metabolism at this node. Indeed, STAT5 inhibition reduced mTORC1 signalling which has been linked to the regulation of glutaminase expression through control of cMyc stability[26]. cMyc is also a direct STAT5 target gene[42]. Interestingly, STAT transcription factors have been linked directly to the expression of glutaminase in monocyte-derived macrophages[43].

Importantly, through the promotion of glutaminolysis, STAT5 signalling supports T-cell function. Akin to STAT5 inhibition, we demonstrate that glutamine withdrawal and

specific inhibition of glutaminolysis impairs human NV CD4+ T-cell function, blocking IL-2 production. We have described the importance of glutamine-derived α-ketoglutarate as an energy substrate in the NV T-cell immune response. Collectively these data support an essential role for glutaminolysis downstream of STAT5 signalling in the human NV T-cell immune response. Importantly, STAT5 signalling has a role in other cell types and disease states, particularly leukaemia[44–46]; as such our work will be pertinent to other systems and pathologies particularly where STAT5 is amenable to targeting.

Collectively, our study demonstrates that the metabolic pathways in human naïve and memory T-cells are distinct. We identify STAT5 as a key node in the early metabolic response in naïve T-cells. The metabolic program orchestrated by STAT5 upon activation facilitates NV T-cell function in a fundamental and causative way allowing the cell to match its bioenergetic capacity to its functional outputs (e.g. cytokine production). Therefore, this study provides insight into the processes accompanying early activation of human CD4+ T-cell subsets, furthering our understanding of the importance of cellular metabolism for T-cell responses.

## Methods

**Human CD4+ T-cell isolation and culture.** Human peripheral blood was collected from healthy, nonfasted individuals into heparinised Vacuettes™ (Greiner Bio-one, Frickenhausen, Germany) and processed within 10 min of collection. All samples were collected with informed written consent and ethical approval was obtained from Wales Research Ethics Committee 6 (13/WA/0190).

Mononuclear cells were initially isolated by layering whole blood (1:1) onto Histopaque (Sigma-Aldrich, Poole, UK) prior to centrifugation at $805 \times g$ for 20 min at room temperature. Mononuclear cells were removed and washed with RPMI 1640 (Life Technologies, Paisley, UK) twice by centrifugation at $515 \times g$. Human CD4+ T-cell subsets were isolated using the autoMACS technique (Miltenyi Biotec). CD4+ NV, EM and CM cells ($1.0 \times 10^6$/mL) were either rested or activated with plate-bound anti-CD3 (2 µg/mL; HIT3a, BioLegend) and free anti-CD28 (20 µg/mL; CD28.2, BioLegend) in phenol red free RPMI (Gibco)+ glutaMAX at 37 °C in 5% CO₂-in-air for 24 h. After 3 h the media was supplemented with 10% fetal calf serum to prevent impaired T-cell activation. For autocrine determination experiments, T-cells were cultured with a common γ chain antibody (1 µg/mL; R&D Systems) or respective isotype control (mouse IgG1, 1 µg/mL; R&D Systems). T-cells were cultured with or without 100 µM STAT5 inhibitor N′-((4-Oxo-4H-chromen-3-yl)methylene) nicotinohydrazide, 1–20 µM Lck inhibitor 4-Amino-5-(4-phenoxyphenyl)-7H-pyrrolo[3,2-d]pyrimidin-7-yl-cyclopentane (Merck Millipore). Oligomycin (100 nM), 6-diazo-5-oxo-L-norleucine (DON; 50 µM), transaminase inhibitor; aminooxyacetic acid (AOA; 0.25 - 1 mM), ATP citrate lyase inhibitor; BMS303141 (1 µM) and dimethyl 2-oxoglutarate (0.3 mM) were purchased from Sigma. Cells were harvested and analysed for flow cytometry and supernatants stored at −20 °C for cytokine analysis. IL-2 was analysed using ELISA as per the manufacturer's instructions (DuoSets; R&D Systems).

**Flow cytometry.** Population purity was measured using anti-CD4 AlexaFluor®647 (mIgG2b, clone OKT4, BioLegend), anti-CD45RA Brilliant Violet 605 (mIgG2b, clone HI100, BioLegend), anti-CD45RO FITC (mIgG2a, clone UCHL1, BioLegend) and anti-CD197 Brilliant Violet 421 (mIgG2a, clone G043H7, BioLegend) antibodies[47]. The percentage purity for individually isolated T-cell populations was consistently >90%.

Cell viability and activation were monitored using nuclear-stain, DRAQ7 (BioStatus) and activation marker, PE-labelled anti-CD69 (mIgG1, FN50, BioLegend), respectively. Mitochondrial content of CD4+ cells was monitored using 2 nM MitoTracker Green (Life Technologies). MFI of CD4+ T-cells with MitoTracker, gated as naive CD45RA⁺CD197⁺, effector memory CD45RA⁻CD197⁻ and central memory CD45RA⁻CD197⁺ to determine mitochondrial content. Total CD4+ T-cells were stained with Brilliant Violet 421 labelled anti-CCR7/CD197 (mIgG2a, clone G043H7, BioLegend) and Brilliant Violet 605 labelled CD45RA (mIgG2b, clone HI100, BioLegend) to identify subset populations. The gating strategy used is identical to Supplementary Fig. 9.

**Metabolic analysis.** Metabolic analysis of CD4+ NV, EM and CM was carried out using the Seahorse Extracellular Flux Analyzer XF°24 (Seahorse Bioscience). Briefly $6.0 \times 10^5$ cells were seeded onto a Cell-Tak (Corning) coated microplate allowing the adhesion of T-cells.

*Baseline*: T-cells were resuspended in XF assay media supplemented with 5.5 mM glucose (Sigma) and 1 mM pyruvate (Sigma). Mitochondrial and glycolytic metabolic parameters were measured simultaneously via OCR (pmoles/min) and

ECAR (mpH/min), respectively (Supplementary Table 1a, b) with use of injections; oligomycin (0.75 µM), FCCP (1 µM) and rotenone and antimycin A (both 1 µM). All chemicals were purchased from Sigma unless stated otherwise. Calculations for individual metabolic parameters can be found in Supplementary Table 1a, b.

*Activation:* To monitor the glycolytic switch upon activation, CD4+ NV, EM and CM cells were resuspended in serum-free XF Assay media supplemented with 11.1 mM glucose and 2 mM L-glutamine (Sigma). ECAR and OCR were measured simultaneously throughout the experiment, i.e. 1 h before activation and 4 h after. T-cells were activated via the multi-injection port with anti-CD3 (0.2 µg/mL; HIT3a, BioLegend) and anti-CD28 (20 µg/mL; CD28.2, BioLegend). A final injection of 2-DG (100 mM; Sigma) was used to arrest glycolysis.

Real-time activation and metabolic flux was monitored via injection of specific inhibitors Akt 1/2 kinase inhibitor (10 µM; Sigma) or STAT5 inhibitor N′-((4-Oxo-4H-chromen-3-yl)methylene)nicotinohydrazide (100 µM; Merck Millipore). Baseline ECAR was measured for 1 h prior to inhibitor injection after which a 40 min period before injection of anti-CD3/CD28.

**Immunoblot.** Freshly isolated NV, EM and CM T-cell lysate proteins were quantified, denatured and separated using SDS-polyacrylamide gel electrophoresis. Polyvinylidene difluoride membranes were probed with antibodies targeting glucose transporter 1 (GLUT1; 12939), hexokinase I (HKI; 2024), hexokinase II (HKII; 2867), glyceraldehyde-3-phosphate dehydrogenase (GAPDH; 5174), phospho-fructokinase (PFK; 8164), pyruvate kinase (PKM2; 4053), lactate dehydrogenase (LDHA; 3582), phospho-STAT5 Tyr694 (9351), total STAT5 (9363), phospho-Akt Thr308 (9275) and Ser473 (9271), phospho-S6 ribosomal protein (Ser235-236; 4858), total S6 ribosomal protein (2217), phospho-p70 S6 kinase (Thr389; 9234) and total p70 S6 kinase (2708). All antibodies were purchased from Cell Signaling (Danvers, MA) and used at a 1:1000 dilution. Protein loading was evaluated and normalised using β-actin (8226; Abcam). Densitometry on nonsaturated immunoblots was measured using ImageJ software (FIJI). Original uncropped immunoblots can be viewed in Supplementary Fig. 10.

**Confocal microscopy.** Isolated CD4+ NV, EM and CM T-cells ($0.1 \times 10^6$ cells) were adhered with Cell-Tak to a Lab-Tek chambered borosilicate coverglass system (ThermoFisher Scientific) and were stained with 20 nM MitoTracker Green. Nuclei were then stained with 5 µM DRAQ5 (BioStatus) and allowed to develop for 15 min before staining the cell membrane with 0.1% CellMask Orange (ThermoFisher Scientific). Live cells were then imaged and captured at ×63 magnification using a laser scanning confocal microscope (Zeiss LSM710). Captured images were analysed using ImageJ (National Institutes of Health, USA).

**Stable isotope tracer analysis (SITA) by GC-MS.** Isolated CD4+ NV, EM and CM were incubated with universally heavy labelled ¹³C glucose (11.1 mM; Cambridge Isotopes) in glucose free RPMI (ThermoFisher Scientific) or ¹³C glutamine (2 mM; Cambridge Isotopes) in glutamine free (ThermoFisher Scientific). T-cells were activated with plate-bound anti-CD3 (2 µg/mL; HIT3a, BioLegend) and free anti-CD28 (20 µg/mL; CD28.2, BioLegend) for a period of either 0.5 or 4 h. Cells were then washed twice with ice-cold PBS and lysed in 80% methanol. Cell extracts were then dried down at 4 °C using a speed-vacuum concentrator.

Cellular metabolites were extracted and analysed by gas chromatography-mass spectrometry (GC-MS) using protocols described previously[48,49]. Briefly, metabolite extracts were derived using N-(tert-butyldimethylsilyl)-N-methyltrifluoroacetamide (MTBSTFA). D-myristic acid (750 ng/sample) was added as an internal standard to metabolite extracts, and metabolite abundance was expressed relative to the internal standard. GC/MS analysis was performed using an Agilent 5975C GC/MS equipped with a DB-5MS + DG (30 m × 250 µm × 0.25 µm) capillary column (Agilent J&W, Santa Clara, CA, USA). For SITA experiments, mass isotopomer distribution was determined using a custom algorithm developed at McGill University[48].

**Extracellular glucose and lactate measurements.** Extracellular glucose and lactate were measured using the glucose assay kit and L-lactate assay kit I (Eton Bioscience) respectively as per the manufacturer's instructions.

**Statistical analysis.** Statistical analysis was performed using GraphPad Prism version 6 (USA). Data are represented as the mean ± or + standard error of the mean (SEM). The one-sample Kolmogorov−Smirnoff test was used to test for normality. Where no substantial deviations from normality were observed it was considered appropriate to use parametric statistics. All experiments have replicate sample sizes of at least $n = 3$ and significant values were taken as $*p \leq 0.05$, $**p \leq 0.01$, $***p \leq 0.001$.

## Reporting summary

Further information on research design is available in the Nature Research Reporting Summary linked to this article.

## Data availability

All relevant data are available from the corresponding author.

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

## Acknowledgements

We thank D. Avizonis and L. Choinière from McGill University Metabolomics Core Facility, J. Rathmell, J. Blagih and D. Elder for useful discussion, S. James for assistance with confocal microscopy, D. Rees for assistance with ImageJ, staff in the Swansea University Joint Clinical Research Facility for phlebotomy, and all blood donors. This work was supported with grants awarded by Saint David's Medical Foundation (SDMF), Life Sciences Research Network Wales (NRN). S.P. was supported by a Wellcome Trust Biomedical Vacation Scholarship. E.E.V. was supported by CRUK (C18281/A19169) and is now supported by a Diabetes UK RD Lawrence Fellowship (17/0005587). D.K.F. is supported by Science Foundation Ireland (13/CDA/2161).

## Author contributions

N.J. performed the majority of experiments; J.G.C., S.P., M.C., S.R.H., S.E.O., N.J.F. and E.E.V. performed experiments and provided intellectual discussion. N.J., E.E.V., D.K.F. and C.A.T. designed the experiments. N.J., E.E.V., D.K.F. and C.A.T. wrote the manuscript. All authors critically revised and approved the manuscript.

## Additional information

**Competing interests:** The authors declare no competing interests.

