## [Peer Review File · Nature Communications]

Reviewers' comments:

Reviewer #1 (Remarks to the Author):

In this paper entitled "An immediate TCR-induced metabolic response in human naïve CD4+ T cells is mediated by Akt and STAT5", Jones and colleagues aim to define the metabolic traits of human CD4+ T cell subpopulations at rest and following in vitro activation. This has been studied in some detail within the human CD8+ T cell compartment, but not yet within human CD4+ T cells. Therefore, this paper makes an interesting contribution to the field, particularly since it appears that the reciprocal CD4+ T cell subsets may have quite different metabolic features to their CD8+ T cell counterparts (especially upon stimulation). In depth metabolic analysis by extracellular flux and stable isotope tracing approaches have been used to probe CD4+ T cell subset metabolism in considerable detail, providing several novel and important insights, for example regarding the role of STAT5 in glutamine metabolism. However I do have some concerns about the experimental approaches employed, presentation of data and conclusions drawn.

Major concerns:

1. When defining metabolic traits of specific T cell subpopulations by extracellular flux or mass spectrometry, it is critical to confirm that highly purified populations are used in all experiments. In this paper, sorted populations of NV, CM and EM CD4+ T cells are used throughout, however information is not provided about which markers were used to sort the CD4+ T cell subpopulations (normally would be a combination of two different markers e.g. CD45RA and CCR7/CD62L) and no data is provided to confirm that the sorting technique used (AutoMACS) generated adequately pure populations.
2. One key novel finding of this work concerns the role of STAT5 signalling downstream of TCR ligation in driving metabolic reprogramming of NV CD4+ T cells. Since it remains controversial whether TCR signalling directly leads to STAT5 phosphorylation in T cells, I think it would be important, in order to definitively prove TCR-induced STAT5 phosphorylation, to rule out a role of autocrine cytokine signalling i.e. IL-2/IL-7 (more likely IL-2) by neutralising this cytokine in culture or blocking its receptor, and then probing for STAT5 phosphorylation following TCR ligation. The experiment performed where IL-2/IL-7 were added to augment STAT5 phosphorylation do not rule out autocrine signalling, and neither does the Lck inhibition experiment.

Other Comments:

Figure 2:

1. Figure 2g is very low resolution, making it difficult to ascertain whether the mitochondrial dye is actually staining structures with mitochondrial morphology. Can these images be improved?
2. In the examples shown in 2g it appears that CM cells have larger areas of mitochondrial staining than EM cells, which is not in agreement with the summarised data presented in 2h. Are these two examples representative?
3. Furthermore, the summarised data presented in 2f and 2h (mito. content EM > CM > NV) are somewhat discordant with the functional data presented in 2d (mito. function CM > EM > NV). Different lymphocyte populations have different capacity to efflux mitochondrial dyes, which can potentially lead to misleading data concerning mitochondrial mass or functionality. This could be controlled for during the assay using inhibitors of this efflux (i.e. of ABC-transporters).
4. In the text describing Figure 2 the sentence "These data demonstrate that CM and EM CD4 T cells do increase oxidation of pyruvate in the TCA cycle relative to NV cells, arguing that CM and EM CD4 T cells sustain elevated OXPHOS using fuels other than glucose" is difficult to understand. Can the authors explain this and justify their conclusion?

Figure 3:

1. It's a little unclear to me what is the justification for the two different approaches to analysis in these experiments (i.e. the 'fold change' immediately following activation and the 'pre-post' analysis, using a 'post' timepoint towards the end of the assay). It also doesn't help that one approach is indicated on the ECAR graph whereas the other is indicated on the OCR graph, but in fact both approaches are used for both ECAR and OCR. Perhaps the authors can explain this a little better for clarity (and either remove the graph annotations or put both on both graphs)?
2. I'm concerned that CM and EM OCR declines steadily during the course of the assay, and furthermore that OCR is then used to calculate ECAR/OCR ratios to define relative glycolytic capacities of the different subsets across the timecourse. Does the declining OCR indicate decreasing viability of these two subsets?

Figure 5:

1. The beta-actin loading control staining looks very variable for the CM cells – is this blot representative of all experiments? Summarised data for a number of experiments would be helpful here.
2. I think that the author's statement that "Inhibition of STAT5 abolished the TCR-induced immediate increase in OCR in NV T cells (Figure 5c)" exaggerates the data, as an increase in OCR following TCR stimulation is still apparent in NV cells.
3. Again I am concerned by the steady decline in OCR of CM and EM T cells over time in these experiments and would like some confirmation that these cells simply do not respond to activation to alter their metabolism because of increased cell death. What other explanation would the authors suggest accounts for the decline in respiration?

Figure 6:

1. In figure 6f there appears to be a significant decrease in not only the phosphorylation of the mTORC1 targets but also of total amount of these proteins in the presence of the STAT5 inhibitor. This makes it difficult to interpret effects on mTORC1 activity per se, yet the authors conclude that it is reduced. Additionally, summarised data would be helpful here.

Statistical Analyses and reproducibility:

Details of statistical tests used are not provided in the methods or figure legends - just a statement that parametric tests in general were used where the data were normally distributed. It is therefore difficult to comment on the appropriateness of these tests. Further information would be helpful. Data are presented as mean +/- standard error of the mean, which is not really appropriate but commonly accepted. In some cases (i.e. for western blot data) no summarised data are provided (indicated above).

Experimental methods are generally very well described with the exception of information on cell sorting as mentioned above.

Reviewer #2 (Remarks to the Author):

In this manuscript, the authors describe an 'immediate' glycolytic change occurring after TCR stimulation in naïve CD4 T cells. The authors implicate Akt and STAT5 driven by the TCR.

This immediate switch has been described by several groups in related cell types (naïve and effector memory CD8s and CD4s). Thus, the enthusiasm for this manuscript would have to be based in how (if at all) naïve CD4s may be different. That is unclear from the manuscript as written.

In Figure 1, one very interesting observation is that naïve T cells express high levels of hexokinase I. The authors mention this but do not follow up? What role does this play, if any, in naïve T cells? Some discussion and experimentation is certainly warranted.

Minor concern: is rotenone/antimycin A the correct control for this experiment? Should 2DG be injected to get baseline levels? How do the authors explain that AA/rotenone reduces ECAR?

In Figure 3, how exactly were these experiments conducted? It appears that the anti-CD3 utilized was solubly injected: is this enough to induce activation? Do these data stay the same if the CD3 antibody is immobilized?

In the text the authors say this is due to TCR, but is it CD28 dependent? Recent papers suggested in CD8 T cells that CD28 is dispensable for immediate glycolysis, but CD4 T cells are likely more dependent on CD28. Would be important to follow up given the authors talk about it in the text.

Figure 4 is a little confusing. Even though the cells immediately switch to glycolysis (minutes), and the authors claim it is due to Akt, there is no apparent increase in pAkt at that timepoint. How do the authors suggest this is occurring? A recent study showed immediate glycolysis in CD8 T cells occurs independent of Akt, so the authors should comment on this a bit and use it as a potential avenue to study the differences, although the dose of Akt inhibitor used (10uM) is very high. The authors should take a genetic approach or use a dose minimally required to inhibit Akt activation in T cells.

In Figure 5, the authors then move to talking about Stat5. I have the same critiques: STAT5 does not appear immediately phosphorylated, so how can it be mediating the switch? STAT5 could certainly be activated directly by the TCR but the authors should address the possibility that the STAT5 signal they're seeing (which really isn't robust until 180 mins) could be due to autocrine IL-2 being secreted as a consequence of TCR stimulation (and thus still inhibited by Lck inhibition). This could be tested using IL-2R blocking antibodies or neutralizing antibodies to IL-2.

The authors should also titrate their dose of STAT5 inhibitor such that is the minimal dose needed, for instance, to reduce IL-7-mediated (for instance) STAT5 to baseline. This may be less than the 100uM they go with.

Likewise, if STAT5 is indeed activating immediate glycolysis and OXPHOS, then the authors should presumably be able to simply inject IL-2 or IL-7 and also see the same phenomenon (as Akt is also activated by these cytokines).

While the idea that STAT5 inhibition may be altering glutaminolysis is interesting, it is surprising that STAT5 inhibition would impair mTORC1 signaling? The authors should dig deeper to identify the mechanism by which this is occurring. Is it a metabolic/nutrient sensing change? Can the mTORC1 signaling be rescued by putting in exogenous glutamine intermediates (aKG, for instance)?

How much of this is dependent (like above) on autocrine IL-2?

Further, how does STAT5 inhibition alter glucose uptake? Is it that STAT5-inhibited cells have more glucose drawn into the pathway, or is this dependent on changes in enzymatic activity?

Finally, some functional consequence of the metabolic reprogramming observed should be shown. While the authors show in the data supplement that STAT5 inhibition impairs cytokine production, this is unsurprising given the role of STAT5 in several T cell gene programs. What would be more important is using some metabolic inhibition of the STAT5, Akt, or TCR-induced changes and link this

to some functional consequences.

Please find below our detailed responses to all Reviewers' comments and concerns marked in blue. *Italicised text* indicates text quoted from the manuscript and underlined text is text added in the revised version.

Reviewer #1 (Remarks to the Author):

In this paper entitled "An immediate TCR-induced metabolic response in human naïve CD4+ T cells is mediated by Akt and STAT5", Jones and colleagues aim to define the metabolic traits of human CD4+ T cell subpopulations at rest and following in vitro activation. This has been studied in some detail within the human CD8+ T cell compartment, but not yet within human CD4+ T cells. Therefore, this paper makes an interesting contribution to the field, particularly since it appears that the reciprocal CD4+ T cell subsets may have quite different metabolic features to their CD8+ T cell counterparts (especially upon stimulation). In depth metabolic analysis by extracellular flux and stable isotope tracing approaches have been used to probe CD4+ T cell subset metabolism in considerable detail, providing several novel and important insights, for example regarding the role of STAT5 in glutamine metabolism. However I do have some concerns about the experimental approaches employed, presentation of data and conclusions drawn.

We thank the Reviewer for their extensive comments regarding our manuscript and are very pleased that they think our manuscript 'makes an interesting contribution to the field' and provides 'several novel and important insights'. We believe these comments have helped us substantiate our conclusions and strengthen our manuscript. Please see below a detailed point-by-point response to the comments and concerns made:

Major concerns:

1. When defining metabolic traits of specific T cell subpopulations by extracellular flux or mass spectrometry, it is critical to confirm that highly purified populations are used in all experiments. In this paper, sorted populations of NV, CM and EM CD4+ T cells are used throughout, however information is not provided about which markers were used to sort the CD4+ T cell subpopulations (normally would be a combination of two different markers e.g. CD45RA and CCR7/CD62L) and no data is provided to confirm that the sorting technique used (AutoMACS) generated adequately pure populations.

We agree with the Reviewer that population purity is critical to any experiments carried out. We assessed population purity every time a T cell isolation was performed. Full details of how population purity was analysed can now be found in the flow cytometry section of the methods. We used CD45RA, CD45RO and CD197 (CCR7) to confirm the purity of our isolated T cell populations. However, we neglected to state that the purity was consistently greater than 90% for each experiment and have now added a statement to clarify this in the methods section:

Flow cytometry

Population purity was measured using standard techniques as described previously with anti-CD4 AlexaFluor®647 (mIgG2b, clone OKT4, eBioscience), anti-CD45RA Brilliant Violet 605 (mIgG2b, clone HI100, BioLegend), anti-CD45RO FITC (mIgG2a, clone UCHL1, BioLegend) and anti-CD197 Brilliant Violet 421 (mIgG2a, clone G043H7, BioLegend) antibodies¹. The percentage purity for individually isolated T cell populations was consistently > 90%.

Reviewer-only Figure 1: Example of CD45RA+ CCR7+ naïve T cell population monitoring.

2. One key novel finding of this work concerns the role of STAT5 signalling downstream of TCR ligation in driving metabolic reprogramming of NV CD4+ T cells. Since it remains controversial whether TCR signalling directly leads to STAT5 phosphorylation in T cells, I think it would be important, in order to definitively prove TCR-induced STAT5 phosphorylation, to rule out a role of autocrine cytokine signalling i.e. IL-2/IL-7 (more likely IL-2) by neutralising this cytokine in culture or blocking its receptor, and then probing for STAT5 phosphorylation following TCR ligation. The experiment performed where IL-2/IL-7 were added to augment STAT5 phosphorylation do not rule out autocrine signalling, and neither does the Lck inhibition experiment.

We thank the Reviewer for this important suggestion and welcome the opportunity to strengthen our conclusion. We agree that it is important to determine whether STAT5 is activated either directly by TCR stimulation or indeed via autocrine signalling. To address this we activated naïve T cells (0.5 and 3 hours) in the presence of a common gamma chain antibody (with respective isotype control).

Initially a pilot experiment was performed to determine the concentration of common gamma chain antibody required to reduced IL-2 mediated STAT5 phosphorylation (Supplementary Figure 7).

We found that STAT5 phosphorylation increased upon TCR activation was unaffected by the presence of the common gamma chain antibody (1 ug/mL) in comparison to the isotype control at both 0.5 and 3 hour time points. These data suggest that the immediate phosphorylation of STAT5 is downstream of the TCR and not mediated by autocrine effects of common gamma chain utilising cytokines such as IL-2 and IL-7. These data have been added to the manuscript in Figure 5m-n and are described in the Results. We thank the Reviewer for this observation and feel it adds substantial weight to our manuscript.

Other Comments:

Figure 2:

1. Figure 2g is very low resolution, making it difficult to ascertain whether the mitochondrial dye is actually staining structures with mitochondrial morphology. Can these images be improved?

Confocal analysis of human T cells remains a challenging technique due to the size of the T cell (roughly 7 μm) in addition to them being non-adherent. We believe that we have now improved the images presented in Figure 2g to the best of our ability and that the updated images are now of quality consistent with previously published work on T cells².

2. In the examples shown in 2g it appears that CM cells have larger areas of mitochondrial staining than EM cells, which is not in agreement with the summarised data presented in 2h. Are these two examples representative?

We thank the Reviewer for highlighting this, we have now updated the figure with representative images of EM and CM T cells mitochondria.

3. Furthermore, the summarised data presented in 2f and 2h (mito. content EM > CM > NV) are somewhat discordant with the functional data presented in 2d (mito. function CM > EM > NV). Different lymphocyte populations have different capacity to efflux mitochondrial dyes, which can potentially lead to misleading data concerning mitochondrial mass or functionality. This could be controlled for during the assay using inhibitors of this efflux (i.e. of ABC-transporters).

The Reviewer raises an interesting point, and we agree that mitochondrial content should not be assumed to translate directly to functionality. Differing mitochondrial morphologies (e.g. mitochondrial networks or spherical entities) can give rise to different efficiencies of mitochondrial respiratory processes largely down to cristae arrangements³. In addition, fuelling of the mitochondria with pyruvate/fatty acids/glutamine/other amino acids can also impact on OXPHOS rates. For these reasons we believe it is important to measure functionality in addition to mitochondria mass and we have done this extensively with the Seahorse assay.

The MitoTracker technique without using ABC-transporter inhibition as a control is a standard within the field (in both human and murine cells)²⁻⁵. We believe this experiment would not further substantiate our conclusions and would not be consistent with the state of the art of the field.

4. In the text describing Figure 2 the sentence “These data demonstrate that CM and EM CD4 T cells do increase oxidation of pyruvate in the TCA cycle relative to NV cells, arguing that CM and EM CD4 T cells sustain elevated OXPHOS using fuels other than glucose” is difficult to understand. Can the authors explain this and justify their conclusion?

We apologise to the Reviewer for the error, the sentence was edited incorrectly. We have corrected the sentence to read: *‘Therefore, these data show that EM and CM CD4+ T cells do not have increased oxidation of pyruvate in the TCA cycle relative to NV cells, arguing that EM and CM CD4+ T cells sustain elevated OXPHOS using fuels other than glucose.’*

Figure 3:

1. It's a little unclear to me what is the justification for the two different approaches to analysis in these experiments (i.e. the 'fold change' immediately following activation and the 'pre-post' analysis, using a 'post' timepoint towards the end of the assay). It also doesn't help that one approach is indicated on the ECAR graph whereas the other is indicated on the OCR graph, but in fact both approaches are used for both ECAR and OCR. Perhaps the authors can explain this a little better for clarity (and either remove the graph annotations or put both on both graphs)?

We thank the Reviewer for highlighting this and we have endeavoured to make this clear in the revised manuscript. The fold change method of analysis has been used to encompass any immediate changes in metabolism (this is especially important in the NV T cell subset, wherein ECAR is readily induced following stimulation). The pre/post method of analysis is used to encompass longer term changes over the duration of the assay (this is important in demonstrating that OCR gradually decreases and ECAR increases in the EM and CM T cell subsets).

We have removed the annotations from both the ECAR and OCR graph in Figure 3a and b to avoid confusion. Supplementary Figure 3a now contains the annotations for clarification. We have added a sentence in the results section to make this clear: 'The differences in ECAR and OCR immediately following stimulation (fold change) and following 257 minutes of stimulation (pre/post) were analysed, the parameters used for these calculations are indicted in Supplementary Figure 3a.'

Some of the figure numbering was incorrect in the original manuscript, which we have now corrected. Finally we have added another sentence to help clarify the reasoning between the two methods of analysis:

'In fact, the increase in glycolysis was greatest in the NV T cells which was significantly greater than in EM CD4+ T cells (Figure 3b, Supplementary Figure 3a-b). The extracellular acidification was significantly increased immediately upon activation for NV and EM T cell subsets, but not CM (Figure 3c). As the assay progressed, ECAR steadily increased for both memory populations. Despite these apparent kinetic differences all three subsets significantly increase their glycolytic rate from baseline post activation (Figure 3d-f).'

2. I'm concerned that CM and EM OCR declines steadily during the course of the assay, and furthermore that OCR is then used to calculate ECAR/OCR ratios to define relative glycolytic capacities of the different subsets across the time course. Does the declining OCR indicate decreasing viability of these two subsets?

We thank the Reviewer for raising this concern. To ensure that the declining OCR does not indicate a decline in viability we have conducted an experiment in which EM T cells have been treated in a homologous fashion to the Seahorse assay followed by a viability test (DRAQ7, flow cytometry). Here we saw no difference in cell death between time zero, the unstimulated 4 hour control and the stimulated 4 hour control.

Reviewer-only Figure 2: Viability of EM T cells assessed using DRAQ7 for freshly isolated, unstimulated and stimulated with anti-CD3/anti-CD28 for 4 hours.

We have also confirmed that even after 24 hours of activation, there is very little cell death amongst all T cell subsets and that death is not induced by STAT5 inhibition (Supplementary Figure 9).

With regards to the reason for the gradual decline in OCR in the EM and CM subsets, we hypothesise that they are gradually switching their metabolism towards glycolysis at the expense of OXPHOS as time progresses (a mechanism that could include HKII translocation to the mitochondria⁶). This is represented in inversely correlated ECAR and OCR values. Both EM and CM T cells have the glycolytic machinery to engage a largely glycolytic metabolism (Figure 1d) whereas the NV T cells are incapable and therefore still rely heavily on OXPHOS.

We have further highlighted this in the discussion: *‘Interestingly we observed an inverse relationship between ECAR and OCR in the EM and CM T cells. Upon activation, both EM and CM T cells gradually increased their ECAR and reduced their OCR over time. This phenomenon could reflect the translocation of HKII toward the mitochondria upon stimulation thus shifting metabolism towards glycolysis as has been observed in other cells^{7,8}. The initial burst of glycolysis observed in NV T cells could be explained by elevated levels of mitochondria-bound HKI, which has been linked previously to increased glycolysis in murine memory T cells³².’*

Figure 5:

1. The beta-actin loading control staining looks very variable for the CM cells – is this blot representative of all experiments? Summarised data for a number of experiments would be helpful here.

We have addressed the issue of the beta actin for the CM T cell subsets and have now added the summarised data to the figure.

2. I think that the author’s statement that “Inhibition of STAT5 abolished the TCR-induced immediate increase in OCR in NV T cells (Figure 5c)” exaggerates the data, as an increase in OCR following TCR stimulation is still apparent in NV cells.

We agree with the Reviewer’s comment and have amended the sentence to read, *‘Inhibition of STAT5 reduced the TCR-induced immediate increase in OCR in NV T cells (Figure 5c)’*.

3. Again I am concerned by the steady decline in OCR of CM and EM T cells over time in these experiments and would like some confirmation that these cells simply do not respond to activation to alter their metabolism because of increased cell death. What other explanation would the authors suggest accounts for the decline in respiration?

We have addressed this issue in full in Figure 3 point 2 above.

Figure 6:

1. In figure 6f there appears to be a significant decrease in not only the phosphorylation of the mTORC1 targets but also of total amount of these proteins in the presence of the STAT5 inhibitor. This makes it difficult to interpret effects on mTORC1 activity per se, yet the authors conclude that it is reduced. Additionally, summarised data would be helpful here.

We have now added the summarised data to the manuscript in Figure 6f. The summarised data demonstrate that phosphorylation of mTORC1 targets remain reduced upon STAT5 inhibition when normalised to total protein. Regardless, STAT5 inhibition clearly affects the activity of the mTORC1 pathway and it may be doing this by modulating the activity of mTORC1 itself and/or the protein stability of mTORC1 targets. To reflect this, we have also added a statement in the discussion: 'This reduction in mTORC1 signalling upon STAT5 inhibition could reflect decreased S6 and p70S6K phosphorylation and/or mTORC1 target protein stability.'

Statistical Analyses and reproducibility:

Details of statistical tests used are not provided in the methods or figure legends - just a statement that parametric tests in general were used where the data were normally distributed. It is therefore difficult to comment on the appropriateness of these tests. Further information would be helpful. Data are presented as mean +/- standard error of the mean, which is not really appropriate but commonly accepted. In some cases (i.e. for western blot data) no summarised data are provided (indicated above). Experimental methods are generally very well described with the exception of information on cell sorting as mentioned above.

We thank the Reviewer for highlighting the missing statistical detail and have now added this to all figure legends. For example the Figure 1 legend now reads:

'Figure 1. Quiescent effector and central memory populations are more metabolically active than naïve T cells. (a) Glycolytic stress profile of NV, EM and CM T cells by measuring ECAR before and following injections of oligomycin (0.75 μ M), FCCP (1 μ M) and antimycin A and rotenone (1 μ M) at the time points indicated. Basal (b) and maximal ECAR (c) in NV, EM and CM T cells. (d) Representative immunoblots from two different donors per cell type for GLUT1, HKI HKII, GAPDH, PFKP, PKM2 and LDHA and β -actin. Respective densitometry normalized to β -actin is shown. Data are either representative of five independent experiments (a-c) or three-four experiments (d). Statistical analysis was performed using a non matching one-way ANOVA with Tukey's multiple comparison test (b-d). For non-parametric data, a Kruskal-Wallis test with Dunn's multiple comparisons test was used. Data expressed as mean \pm SEM; * $p \leq 0.05$, ** $p \leq 0.01$, * $p \leq 0.001$.'**

Furthermore, we have added the missing summarised data for all western blotting figures. Finally, we have commented on the percentage purity of isolated populations in the flow cytometry section, where the method of purity analysis can also be found.

Reviewer #2 (Remarks to the Author):

In this manuscript, the authors describe an 'immediate' glycolytic change occurring after TCR stimulation in naïve CD4 T cells. The authors implicate Akt and STAT5 driven by the TCR.

We would like to thank the Reviewer for their insightful comments on the manuscript and feel it is much improved following our revisions. Please see below detailed responses to all the comments.

This immediate switch has been described by several groups in related cell types (naïve and effector memory CD8s and CD4s). Thus, the enthusiasm for this manuscript would have to be based in how (if at all) naïve CD4s may be different. That is unclear from the manuscript as written.

We thank the reviewer for highlighting that we had not effectively communicated the novel contributions to the field that our manuscript makes with regards to the differences between CD4+ T cell subset metabolism in comparison to that of CD8+ T cells.

Firstly, it is important to stress that to date the literature is lacking an extensive study into human CD4+ T cell naïve, effector and central memory metabolism. Specifically, this is the first time that the immediate kinetics of human CD4+ T cell activation has been investigated. A previous study has investigated human CD8+ T cell subsets⁹ but not CD4+ T cells. We believe these novel findings are an extremely important contribution to the field. Indeed, in order to make use of novel therapeutic interventions that target metabolic pathways (such as in autoimmune disease and cancer) and understand how they impact T cell function, CD4+ T cell subset metabolism must first be characterised and understood. In this context it is also important to understand how CD4+ T cell metabolism differs from that of CD8+ T cells.

The data we present demonstrates that human CD4+ T cell subsets behave very differently from their CD8+ counterparts. For example, NV CD4+ T cells respond vigorously to stimulation (anti-CD3/anti-CD28) in comparison to NV CD8+ T cells. The NV CD4+ T cells engage glycolysis at a higher rate than CD4+ EM/CM T cells, whereas CD8+ EM T cells engage glycolysis at a higher rate than the NV CD8+ T cells. In addition, we demonstrate clear differences in the glycolytic enzyme expression profile during quiescence between CD4+ T cells subsets, no differences were reported in CD8+ T cell subsets by *Gubser et al.*

We also demonstrate that CD4+ NV T cells respond metabolically to stimulation (anti-CD3/anti-CD28) similarly to CD8+ EM T cells. Indeed, we find that it is only the NV CD4+ T cell compartment that increase their oxygen consumption upon stimulation, whereas EM and CM T cells do not. We are also the first to measure TCA cycle intermediates in human CD4+ T cell subpopulations (importantly at early time points) and report differences between the subpopulations.

Furthermore, in an experiment suggested by the Reviewer, we have determined that the immediate glycolytic and oxidative switch in activated naïve CD4+ T cells is dependent on both CD3 and CD28. A previous study in murine NV CD8+ and CD8+ previously activated T cells¹⁰ reported that CD28 is dispensable. Here we conclude that this is not that case, demonstrating that human CD4+ naïve T cells have differential requirements for activation in comparison to murine naïve CD8+ T cells.

Finally, an equally important finding of our manuscript is the description of the novel metabolic mechanism underlying NV CD4+ activation, implicating the signalling molecules Akt and STAT5. A

role for STAT5 in T cell metabolism has not been described previously and we believe the finding that it regulates glutaminolysis through mTORC1 will be of significant interest to the field.

In order to ensure we have effectively communicated these novel contributions we have made changes to the manuscript text, these are detailed below (underlined text indicates additions to the manuscript):

1. Abstract: We have made changes in the abstract to emphasise the novelty in describing the metabolic differences in CD4+ T cell subsets in comparison to CD8+ T cells.

Page 2, line 2:

'However, in comparison to human CD8+ T cells, little is known about the metabolic pathways utilized by different human CD4+ T cell subsets and how these compare to their CD8+ counterparts.'

Page 2, line 7:

'Despite having low resting metabolic rates, NV CD4+ T cells respond to TCR (T cell receptor)-stimulation with robust and immediate increases in glycolysis and OXPHOS in comparison to EM and CM CD4+ T cells, demonstrating clear metabolic differences to their CD8+ counterparts.'

2. Introduction: We have included a section to highlight the novelty of our manuscript with regards to the differences between CD4 T cell subset metabolism in comparison to that of CD8 T cells.

Page 4, paragraph 2:

'Herein, we characterise the metabolic phenotypes of human CD4+ NV, EM and CM T cells and investigate the contributions of the signalling molecules Akt and signal transducer and activator of transcription 5 (STAT5) towards early T cell metabolism upon activation. Importantly, we uncover various metabolic differences that are not homologous to human CD8+ T cell subsets⁸. Here we show that NV CD4+ T cells differ from their CD8+ counterparts by rapidly engaging glycolysis and oxygen consumption upon stimulation. We find that NV CD4+ T cells are heavily reliant on their mitochondria upon activation utilising glutamine for processes such as reductive carboxylation. This is linked to reduced expression of key glycolytic enzymes in NV T cells, compared to EM and CM T cells.'

3. Discussion: The discussion contains extensive discussion of the differences in CD4+ T cell subset metabolism in comparison to their CD8+ counterparts (particularly paragraph 3). We have made an addition in order to discuss the interesting experiment suggested by the Reviewer with regards to CD4+ T cell dependency on CD28.

Page 17, paragraph 2:

'In contrast to CM and EM CD4+ T cells, and indeed NV CD8+ T cells, NV CD4+ T cells do engage an immediate metabolic response following TCR/CD28 stimulation that involves increased rates of glycolysis and OXPHOS. While an immediate glycolytic response following TCR stimulation has been observed in murine and human CD8+ T cells, this is the first time this has been described in NV CD4+ T cells. Interestingly, we found that increased glycolysis levels in activated NV T cells were dependent on CD28 ligation, whereby in murine CD8+ NV and previously activated T cells increased glycolysis levels were independent of CD28 in the immediate term²⁹.'

In Figure 1, one very interesting observation is that naïve T cells express high levels of hexokinase I. The authors mention this but do not follow up? What role does this play, if any, in naïve T cells? Some discussion and experimentation is certainly warranted.

We agree with the Reviewer that the observation that HKI appears to be elevated in NV T cells compared to EM and CM T cells is an interesting observation and certainly warrants future investigation. Indeed, we have plans to follow this up in detail, however, we feel that it is beyond the scope of this current manuscript to pursue this further here. We agree with the reviewer that further discussion of this is warranted in the current manuscript and have added a paragraph to the discussion in the revised version of the manuscript.

'Interestingly we observed an inverse relationship between ECAR and OCR in the EM and CM T cells. Upon activation, both EM and CM T cells gradually increased their ECAR and reduced their OCR over time. This phenomenon could reflect the translocation of HKII toward the mitochondria upon stimulation thus shifting metabolism towards glycolysis and away from OXPHOS³¹. The initial burst of glycolysis observed in NV T cells could be explained by elevated levels of mitochondria-bound HKI, which has been linked previously to increased glycolysis in murine memory T cells³².'

Minor concern: is rotenone/antimycin A the correct control for this experiment? Should 2DG be injected to get baseline levels? How do the authors explain that AA/rotenone reduces ECAR?

Rotenone/antimycin A has been used previously as a control for this type of assay¹¹. Rotenone and antimycin A inhibit complexes I and III respectively. The reason we believe for the decline in ECAR upon rotenone/antimycin A injection would be the removal of CO₂ generated from the TCA cycle which can affect acidification rates¹².

In Figure 3, how exactly were these experiments conducted? It appears that the anti-CD3 utilized was solubly injected: is this enough to induce activation? Do these data stay the same if the CD3 antibody is immobilized?

Both antibodies (anti-CD3 and anti-CD28) were soluble when injected. This approach has been utilised by other groups to activate both murine and human T cell subsets^{9,10}. In the assay protocol we use, the T cells need to be adhered to the Seahorse plate prior to the assay for the assay to work, this is achieved using Cell-Tak. Unfortunately immobilising the CD3 antibody is not compatible with the Cell-Tak protocol.

In the text the authors say this is due to TCR, but is it CD28 dependent? Recent papers suggested in CD8 T cells that CD28 is dispensable for immediate glycolysis, but CD4 T cells are likely more dependent on CD28. Would be important to follow up given the authors talk about it in the text.

This is an interesting point raised by the Reviewer and we have addressed it experimentally. We find that CD4+ T cell activation is dependent on CD28. Interestingly, we found that both anti-CD3 and anti-CD28 were required for the immediate increase in oxygen consumption. However CD28 alone induced a switch in ECAR (albeit to a lesser extent than anti-CD3 and anti-CD28 combined). We have added our findings to supplementary Figure 3 and have added text in the results section accordingly:

'We also discovered that the immediate increase in ECAR in NV cells is dependent on CD28 (however the combined effect of both CD28 and CD3 remains greater), however the immediate OCR switch was entirely dependent on both CD3 and CD28 stimulation (Supplementary Figure 3d-e).'

We have also added to the discussion on this interesting point:

'In contrast to CM and EM CD4+ T cells, NV CD4+ T cells do engage an immediate metabolic response

following TCR/CD28 stimulation that involves increased rates of glycolysis and OXPHOS. While an immediate glycolytic response following TCR stimulation has been observed in murine and human CD8+ T cells, this is the first time this has been described in NV CD4+ T cells. Interestingly, we found that increased glycolysis levels in activated NV T cells were dependent on CD28 ligation, whereas in murine CD8+ NV and previously activated T cells increased glycolysis levels were independent of CD28 in the immediate term²⁹.

Figure 4 is a little confusing. Even though the cells immediately switch to glycolysis (minutes), and the authors claim it is due to Akt, there is no apparent increase in pAkt at that timepoint. How do the authors suggest this is occurring? A recent study showed immediate glycolysis in CD8 T cells occurs independent of Akt, so the authors should comment on this a bit and use it as a potential avenue to study the differences, although the dose of Akt inhibitor used (10uM) is very high. The authors should take a genetic approach or use a dose minimally required to inhibit Akt activation in T cells.

The Reviewer raises important points and we have addressed them experimentally. We originally selected the 10 μ M concentration of the Akt1/2 inhibitor as it has been used previously to study metabolism in human CD8+ T cell subsets^{9,13}. It is important to note that 10 μ M treatment of the Akt1/2 inhibitor does not affect OCR in NV T cells (Figure 4e). This suggests that the action of the inhibitor on metabolism is specific and lends strength to the conclusion that 10 μ M is an appropriate concentration to use as it does not have a global detrimental effect on T cells (e.g. on viability).

We have conducted an Akt1/2 inhibitor titration experiment to confirm that the effect on ECAR we observe in NV CD4+ T cells upon Akt inhibition is dose dependent and apparent at lower doses of the inhibitor (**Reviewer-only Figure 3**). These data demonstrate that reduced ECAR is apparent at 1 μ M, suggesting that the effect is specific to Akt inhibition and unlikely to be caused by off-target effects of the inhibitor.

Reviewer-only Figure 3: Titration of the Akt1/2 inhibitor (Akt1/2i). (a) ECAR and (b) OCR in NV T cells upon treatment with Akt1/2i (10, 1, 0.1 and 0.01 μ M) and stimulation with anti-CD3 and anti-CD28 (0.2 and 20 μ g/mL respectively).

Importantly, we have conducted an additional set of experiments in NV T cells whereby we demonstrate that after 15 and 30 minutes of activation both STAT5 and Akt are clearly phosphorylated (Supplementary Figure 4a). We have also included an experiment in the revised version of the manuscript that demonstrates that the Akt1/2 and STAT5 inhibitors inhibit Akt and STAT5 respectively at the concentrations used in the manuscript (Supplementary Figure 4b). In this experiment NV T cells were activated for 30 minutes. This experiment demonstrates a clear

reduction in the levels of phosphorylated STAT5 and Akt in the presence of their respective inhibitors. This obvious difference in phosphorylation in the absence vs the presence of the inhibitor is consistent with the clear difference in ECAR observed in the Seahorse experiments. This experiment also demonstrates that there is already a robust window in both Akt and STAT5 phosphorylation 30 minutes after activation (when comparing absence vs presence of inhibitor). This is consistent with the Seahorse data in Figures 4d and 5c and f.

Additions have been made to the Results section of the manuscript to accommodate this:

'TCR stimulation induced the phosphorylation of Akt on threonine 308 and serine 473, with more robust activation observed in NV T cells (Figure 4a-c). An experiment to demonstrate that robust activation of Akt was apparent at earlier timepoints is presented in Supplementary Figure 4a. To investigate whether Akt was required for the immediate glycolytic response in NV CD4+ T cells the allosteric inhibitor Akti-1/2 was used to inhibit Akt activity prior to TCR triggering (efficacy of the inhibitor at the concentration used here is demonstrated in Supplementary Figure 4b).'

'An interesting observation was that TCR-stimulation induced the phosphorylation of STAT5 on tyrosine 694 in NV, CM and EM CD4+ T cells (Figure 5a). An experiment to demonstrate that robust activation of STAT5 was apparent at earlier timepoints is presented in Supplementary Figure 4a.'

'A commonly used STAT5 inhibitor N'-((4-Oxo-4H-chromen-3-yl)methylene)nicotinohydrazide (STAT5i) was optimised (Supplementary Figure 5b and 6) and used (Figure 5c-h) to investigate the role for STAT5 in TCR-induced cellular metabolism¹⁴.'

We thank the reviewer for raising the point about immediate glycolysis in CD8+ T cells occurring independent of Akt. We have added a discussion point about this to the revised manuscript.

'Additionally, the data herein demonstrate that the immediate glycolytic response in NV CD4+ T cells is completely dependent on Akt activity as is shown to be the case in CD8+ EM T cells⁹. This is in contrast to activated murine NV CD8+ T cells where the early glycolytic switch is mediated by pyruvate dehydrogenase kinase 1 (PDHK1) independent of Akt¹⁰.'

In Figure 5, the authors then move to talking about Stat5. I have the same critiques: STAT5 does not appear immediately phosphorylated, so how can it be mediating the switch? STAT5 could certainly be activated directly by the TCR but the authors should address the possibility that the STAT5 signal they're seeing (which really isn't robust until 180 mins) could be due to autocrine IL-2 being secreted as a consequence of TCR stimulation (and thus still inhibited by Lck inhibition). This could be tested using IL-2R blocking antibodies or neutralizing antibodies to IL-2.

The Reviewer raises an important point in identifying whether STAT5 phosphorylation is due to TCR stimulation or autocrine signalling via IL-2 secretion. We have addressed this concern in full above (please see Reviewer 1, point 2).

The remainder of this comment has been addressed in the response to the last point above. With regards to STAT5 phosphorylation not appearing to be immediately phosphorylated, we feel the representative blots and summarised data in Figure 5a do show an increase above baseline in STAT5 phosphorylation at 15 minutes. This is now also very clearly demonstrated in Supplementary Figure 4a. In additional experimental data in Figure 5m we also show that STAT5 phosphorylation is clearly induced following 30 minutes of activation. This is consistent with the Seahorse data in Figure 5.

The authors should also titrate their dose of STAT5 inhibitor such that is the minimal dose needed, for instance, to reduce IL-7-mediated (for instance) STAT5 to baseline. This may be less than the 100uM they go with.

We have used a previously published concentration of the STAT5 inhibitor (Sallusto lab) used on human T cells¹⁴. In addition, Supplementary Figure 6 demonstrates that NV cell ECAR responds to STAT5 inhibition in a dose dependent manner. As the effect is dose-dependent it suggests this is unlikely due to an off-target effect of the inhibitor. At this dose the inhibitor does not affect cell viability (Supplementary Figure 9) and does not affect EM or CM metabolism (Figure 5), further suggesting the dose used does not have global off target effects.

Likewise, if STAT5 is indeed activating immediate glycolysis and OXPHOS, then the authors should presumably be able to simply inject IL-2 or IL-7 and also see the same phenomenon (as Akt is also activated by these cytokines).

We agree with the Reviewer that it would indeed be interesting to look at the metabolic response upon stimulation with IL-2 and IL-7 and we have addressed this experimentally. However, we found that neither IL-2 nor IL-7 affected ECAR or OCR in CD4+ NV T cells. It has been reported previously in both murine and human T cells that TCR/CD28 stimulation engages a substantial immediate metabolic response. We believe that STAT5 is required but not sufficient to induce the changes in ECAR/OCR that we observe in CD4+ NV T cells. We predict that there are multiple other essential signals downstream of TCR/CD28 stimulation that are not present upon IL-2 and IL-7 stimulation. A previous study has examined glucose uptake upon IL-7 stimulation in murine T cells¹⁵. Whilst IL-7 did increase glucose uptake in an Akt-dependent manner, this was observed after overnight culture.

These data are an important addition to the manuscript and lend strength and context to our conclusions. We have added the findings as Figure 5k and l and included the following text in the results section:

'As IL-2 and IL-7 are known inducers of Akt and STAT5^{15,16}, we next investigated whether the cytokines would individually induce an increase in OCR or ECAR (Figure 5k-l). However, unlike TCR stimulation, exposure to IL-2 or IL-7 did not affect the immediate metabolic rates of NV T cells.'

We have also discussed these data in the discussion: 'In contrast to TCR stimulation, we did not observe any metabolic changes in NV T cells stimulated with IL-2 or IL-7; this could reflect the fact that NV T cells are constantly bathed in cytokines within the lymph nodes, preventing the occurrence of unnecessary activation. Therefore, we can conclude that STAT5 is required but not sufficient for NV T cell activation.'

While the idea that STAT5 inhibition may be altering glutaminolysis is interesting, it is surprising that STAT5 inhibition would impair mTORC1 signaling? The authors should dig deeper to identify the mechanism by which this is occurring. Is it a metabolic/nutrient sensing change? Can the mTORC1 signaling be rescued by putting in exogenous glutamine intermediates (αKG, for instance)?

We agree with the reviewer that this is an extremely interesting finding and we have further explored the dependence of NV CD4+ T cells on glutaminolysis in new Figure 8 in the revised version of the manuscript. We discuss this in detail in response to the final comment of the Reviewer below.

There is extensive cross talk between the STAT5 and mTORC1 pathways in immune cells so it does not seem surprising that STAT5 inhibition would impair mTORC1 signalling¹⁷. The exact mechanistic link between STAT5 and mTORC1 will indeed be interesting to investigate and we plan to follow this up, however, we feel that this would be beyond the scope of the current manuscript.

How much of this is dependent (like above) on autocrine IL-2?

This point has been addressed above.

Further, how does STAT5 inhibition alter glucose uptake? Is it that STAT5-inhibited cells have more glucose drawn into the pathway, or is this dependent on changes in enzymatic activity?

We have addressed this experimentally by investigating the levels of glucose in the media between vehicle and STAT5 inhibitor treated NV T cells. Levels of glucose are higher in supernatant from NV T cells treated with the STAT5 inhibitor in comparison to the vehicle control. These data are presented in Figure 5j. These data suggest that STAT5 inhibition affects glucose uptake and thereby reduces the availability of glucose for glycolysis. We have updated the method, results section and corresponding figure legend with details of the glucose assay.

Finally, some functional consequence of the metabolic reprogramming observed should be shown. While the authors show in the data supplement that STAT5 inhibition impairs cytokine production, this is unsurprising given the role of STAT5 in several T cell gene programs. What would be more important is using some metabolic inhibition of the STAT5, Akt, or TCR-induced changes and link this to some functional consequences.

The Reviewer makes an excellent point regarding the functional consequence of impaired glutaminolysis. In order to investigate this, we deprived activated NV T cells of glutamine alongside culturing them with a specific glutaminase inhibitor, CB-839, and subsequently measured IL-2 production, cell viability and cell size. Here we found that both glutamine restriction and CB-839 treatment impaired IL-2 production in NV T cells. We have added this to the manuscript as Figure 8, with details included in the results section:

'Glutaminolysis inhibition compromises IL-2 production

As STAT5 inhibition disrupted glutamine anaplerosis into the TCA cycle, we next wanted to understand the functional consequence of impaired glutaminolysis in human NV T cells. In order to address this we employed two approaches; the first utilised a glutaminase inhibitor CB-839 whilst the second investigated glutamine deprivation. NV T cells activated for 24 hours in the presence of CB-839 or the absence of glutamine (-) reduced IL-2 production (Figure 8a). Glutamine restriction had a slight impact on NV T cell viability and cell size after 24 hours; whereas the cell viability and size of CB-839 treated NV T cells did not differ from the vehicle control (Figure 8b-c). These differences are likely to reflect that glutamine has additional functions in lymphocytes apart from fuelling glutaminolysis, such as for signal transduction^{18,19}.

Discussion:

'A recent study showed that glutamine restriction favoured the generation of CD4+ T cells with high expression of the Treg transcription factor FOXP3 suggesting that glutamine metabolism is important in the differentiation of effector versus regulatory human CD4+ T cell subsets²⁰. Here we show that specific glutaminase inhibition via CB-839 or complete removal of glutamine from the media

impacted negatively on IL-2 production from human NV T cells. Whilst glutamine withdrawal affected NV CD4+ T cell size and viability, inhibition of glutaminase using CB-839 did not. This is consistent with the finding that STAT5 inhibition also did not impact viability.'

We are grateful to the Reviewer for these suggestions as we feel the addition of Figure 8 greatly improves the impact of our manuscript and substantiates our conclusions.

- 1 Jones, N. *et al.* Bioenergetic analysis of human peripheral blood mononuclear cells. *Clin Exp Immunol* **182**, 69-80, doi:10.1111/cei.12662 (2015).
- 2 van der Windt, G. J. *et al.* CD8 memory T cells have a bioenergetic advantage that underlies their rapid recall ability. *Proc Natl Acad Sci U S A* **110**, 14336-14341, doi:10.1073/pnas.1221740110 (2013).
- 3 Buck, M. D. *et al.* Mitochondrial Dynamics Controls T Cell Fate through Metabolic Programming. *Cell* **166**, 63-76, doi:10.1016/j.cell.2016.05.035 (2016).
- 4 Quintana, A. *et al.* T cell activation requires mitochondrial translocation to the immunological synapse. *Proc Natl Acad Sci U S A* **104**, 14418-14423, doi:10.1073/pnas.0703126104 (2007).
- 5 Klein Geltink, R. I. *et al.* Mitochondrial Priming by CD28. *Cell* **171**, 385-397 e311, doi:10.1016/j.cell.2017.08.018 (2017).
- 6 John, S., Weiss, J. N. & Ribalet, B. Subcellular localization of hexokinases I and II directs the metabolic fate of glucose. *PLoS One* **6**, e17674, doi:10.1371/journal.pone.0017674 (2011).
- 7 Calmettes, G., John, S. A., Weiss, J. N. & Ribalet, B. Hexokinase-mitochondrial interactions regulate glucose metabolism differentially in adult and neonatal cardiac myocytes. *J Gen Physiol* **142**, 425-436, doi:10.1085/jgp.201310968 (2013).
- 8 Everts, B. *et al.* TLR-driven early glycolytic reprogramming via the kinases TBK1-IKKvarepsilon supports the anabolic demands of dendritic cell activation. *Nat Immunol* **15**, 323-332, doi:10.1038/ni.2833 (2014).
- 9 Gubser, P. M. *et al.* Rapid effector function of memory CD8+ T cells requires an immediate-early glycolytic switch. *Nat Immunol* **14**, 1064-1072, doi:10.1038/ni.2687 (2013).
- 10 Menk, A. V. *et al.* Early TCR Signaling Induces Rapid Aerobic Glycolysis Enabling Distinct Acute T Cell Effector Functions. *Cell Rep* **22**, 1509-1521, doi:10.1016/j.celrep.2018.01.040 (2018).
- 11 Kolev, M. *et al.* Complement Regulates Nutrient Influx and Metabolic Reprogramming during Th1 Cell Responses. *Immunity* **42**, 1033-1047, doi:10.1016/j.immuni.2015.05.024 (2015).
- 12 Mookerjee, S. A., Goncalves, R. L. S., Gerencser, A. A., Nicholls, D. G. & Brand, M. D. The contributions of respiration and glycolysis to extracellular acid production. *Biochim Biophys Acta* **1847**, 171-181, doi:10.1016/j.bbabbio.2014.10.005 (2015).
- 13 Bantug, G. R. *et al.* Mitochondria-Endoplasmic Reticulum Contact Sites Function as Immunometabolic Hubs that Orchestrate the Rapid Recall Response of Memory CD8(+) T Cells. *Immunity* **48**, 542-555 e546, doi:10.1016/j.immuni.2018.02.012 (2018).
- 14 Zielinski, C. E. *et al.* Pathogen-induced human TH17 cells produce IFN-gamma or IL-10 and are regulated by IL-1beta. *Nature* **484**, 514-518, doi:10.1038/nature10957 (2012).

- 15 Wofford, J. A., Wieman, H. L., Jacobs, S. R., Zhao, Y. & Rathmell, J. C. IL-7 promotes Glut1 trafficking and glucose uptake via STAT5-mediated activation of Akt to support T-cell survival. *Blood* **111**, 2101-2111, doi:10.1182/blood-2007-06-096297 (2008).
- 16 Hand, T. W. *et al.* Differential effects of STAT5 and PI3K/AKT signaling on effector and memory CD8 T-cell survival. *Proc Natl Acad Sci U S A* **107**, 16601-16606, doi:10.1073/pnas.1003457107 (2010).
- 17 Saleiro, D. & Platanias, L. C. Intersection of mTOR and STAT signaling in immunity. *Trends Immunol* **36**, 21-29, doi:10.1016/j.it.2014.10.006 (2015).
- 18 Swamy, M. *et al.* Glucose and glutamine fuel protein O-GlcNAcylation to control T cell self-renewal and malignancy. *Nat Immunol* **17**, 712-720, doi:10.1038/ni.3439 (2016).
- 19 Loftus, R. M. *et al.* Amino acid-dependent cMyc expression is essential for NK cell metabolic and functional responses in mice. *Nat Commun* **9**, 2341, doi:10.1038/s41467-018-04719-2 (2018).
- 20 Metzler, B., Gfeller, P. & Guinet, E. Restricting Glutamine or Glutamine-Dependent Purine and Pyrimidine Syntheses Promotes Human T Cells with High FOXP3 Expression and Regulatory Properties. *J Immunol* **196**, 3618-3630, doi:10.4049/jimmunol.1501756 (2016).

Reviewers' comments:

Reviewer #1 (Remarks to the Author):

The authors have responded satisfactorily to almost all of my comments and concerns and have thereby reassured me as to the novel aspects of this research. They have also clarified several issues in the manuscript which I found difficult to interpret. I do think however that the other reviewer raised some valid issues which the authors have not fully addressed, but don't feel it is my place to further comment on that.

Reviewer #2 (Remarks to the Author):

The authors have included a number of experiments addressing my concerns, which provide some interesting results. However, in its current form the manuscript remains descriptive. Much of the more provocative questions were relegated in the response to future work.

The impact of this study seems to stem from the observation that naïve CD4 T cells are somehow different in their signaling requirements to engage metabolic decisions. The fact that no one else has profiled human naïve T helper cells before is notable but the authors provide no mechanistic insight into why they are different.

Further, essentially the only readouts in this study are metabolic. The authors need to equate the metabolic changes to some functional differences in naïve T cells. Does early cytokine production change? Differentiation? Proliferation? What significance does this signaling-metabolic crosstalk have for T cell function?

Response to reviewers' comments:

Reviewer #1 (Remarks to the Author): The authors have responded satisfactorily to almost all of my comments and concerns and have thereby reassured me as to the novel aspects of this research. They have also clarified several issues in the manuscript which I found difficult to interpret. I do think however that the other reviewer raised some valid issues which the authors have not fully addressed, but don't feel it is my place to further comment on that.

We thank the Reviewer for their comments and we are glad that our previous responses satisfied their concerns with regards to the novelty of our research and clarified issues. We hope that our further revised version would resolve any issues they felt were remaining.

Reviewer #2 (Remarks to the Author): The authors have included a number of experiments addressing my concerns, which provide some interesting results. However, in its current form the manuscript remains descriptive. Much of the more provocative questions were relegated in the response to future work.

The impact of this study seems to stem from the observation that naïve CD4 T cells are somehow different in their signaling requirements to engage metabolic decisions. The fact that no one else has profiled human naïve T helper cells before is notable but the authors provide no mechanistic insight into why they are different.

Further, essentially the only readouts in this study are metabolic. The authors need to equate the metabolic changes to some functional differences in naïve T cells. Does early cytokine production change? Differentiation? Proliferation? What significance does this signaling-metabolic crosstalk have for T cell function?

We thank the Reviewer for their careful and constructive review. We feel that their comments have helped us strengthen the manuscript and substantiate our conclusions. We agree with the Reviewer and we were keen to demonstrate that the metabolic phenotype that we describe is linked in a causative way to T cell function.

A key novel message of our manuscript is that STAT5 acts as a central node in naïve CD4+ T cells to control both the expression of important T cell effector genes (e.g cytokines) and to coordinate a metabolic response to allow the cell to make the energy and the building blocks (e.g. amino acids) to actually synthesise these effector molecules. In the revised version of the manuscript we confirm that STAT5 inhibition itself drastically impairs T cell function (Figure 8a). To determine the functional dependence of NV CD4+ T cells on the STAT5 mediated early metabolic program that we have characterised in this study we demonstrate that glutaminolysis (regulated by STAT5) is essential for naïve T cell function (Figure 8b-d). We also show that glutamine carbon withdrawal can be rescued with supplementation of dimethyl 2-oxoglutarate (DMK). DMK is a membrane permeating α -ketoglutarate analog and can be used to supply carbon to the TCA cycle. The addition of DMK restored IL-2 production in glutamine-deprived conditions (Figure 8f), demonstrating the key role for this pathway in naïve T cell function. These data therefore provide a causative link between STAT5-controlled metabolism and T cell function and demonstrate how the cell matches its bioenergetic and biosynthetic capacity to its functional outputs (cytokine production). We chose to measure IL-2 production as a functional output due to NV T cells being restricted in their cytokine production profiles¹.

In the revised version of the manuscript we include several additional novel findings. These include evidence that human naïve CD4+ T cells generate glutamine-derived aspartate (Supplementary Figure 8b). The dependence of naïve CD4+ T cells on STAT5 signalling for the production of non-essential amino acids from glucose and glutamine is also a novel finding (Figure 6c, e). We also now demonstrate that alpha-ketoglutarate is sufficient to rescue the function of glutamine deprived T cells because it drives the TCA cycle for energy production (using the ATP synthase inhibitor oligomycin); α KG fuelled biosynthesis appears to be less important for T cell IL-2 production (data using the ACLY inhibitor BMS303141) (Figure 8g-h).

1. Ohshima Y, Yang LP, Avicé MN, et al. Naive human CD4+ T cells are a major source of lymphotoxin alpha. *J Immunol.* 1999;162(7):3790-3794.

REVIEWERS' COMMENTS:

Reviewer #1 (Remarks to the Author):

The authors have now added additional experimental evidence to link the specific metabolic features of naive CD4+ T cells that they describe to a key immune function of these cells (namely IL-2 production). The new data support their overall conclusions that STAT5 is a key signaling intermediary driving the function of these cells, and furthermore that STAT5-mediated alterations in glutamine uptake and metabolism critically support IL-2 production. The requirement for glutamine and glutaminolysis is now confirmed by depletion and 'add-back' experiments. I think these new data add to the significance of this work and would continue to support its publication in this journal.

Reviewer #2 (Remarks to the Author):

The authors have satisfied my concerns.

Reviewer #3 (Remarks to the Author):

The revised manuscript addresses my major concerns. Although I am not fully convinced by their claims derived from these results, this remains to be an interesting paper to be published.

REVIEWERS' COMMENTS:

Reviewer #1 (Remarks to the Author):

The authors have now added additional experimental evidence to link the specific metabolic features of naive CD4+ T cells that they describe to a key immune function of these cells (namely IL-2 production). The new data support their overall conclusions that STAT5 is a key signaling intermediary driving the function of these cells, and furthermore that STAT5-mediated alterations in glutamine uptake and metabolism critically support IL-2 production. The requirement for glutamine and glutaminolysis is now confirmed by depletion and 'add-back' experiments. I think these new data add to the significance of this work and would continue to support its publication in this journal.

Thank you for these comments

Reviewer #2 (Remarks to the Author):

The authors have satisfied my concerns.

Thank you

Reviewer #3 (Remarks to the Author):

The revised manuscript addresses my major concerns. Although I am not fully convinced by their claims derived from these results, this remains to be an interesting paper to be published.

Thank you for these comments